# A translational program that suppresses metabolism to shield the genome

Nathan C. Balukoff [1,2,7], J. J. David Ho[1,2,3,7], Phaedra R. Theodoridis[1,2], Miling Wang[1,2], Michael Bokros[1,2], Lis M. Llanio[1], Jonathan R. Krieger[4,6], Jonathan H. Schatz [2,3] & Stephen Lee [1,2,5 ✉]

Translatome reprogramming is a primary determinant of protein levels during stimuli adaptation. This raises the question: what are the translatome remodelers that reprogram protein output to activate biochemical adaptations. Here, we identify a translational pathway that represses metabolism to safeguard genome integrity. A system-wide MATRIX survey identified the ancient eIF5A as a pH-regulated translation factor that responds to fermentation-induced acidosis. TMT-pulse-SILAC analysis identified several pH-dependent proteins, including the mTORC1 suppressor Tsc2 and the longevity regulator Sirt1. Sirt1 operates as a pH-sensor that deacetylates nuclear eIF5A during anaerobiosis, enabling the cytoplasmic export of eIF5A/Tsc2 mRNA complexes for translational engagement. Tsc2 induction inhibits mTORC1 to suppress cellular metabolism and prevent acidosis-induced DNA damage. Depletion of eIF5A or Tsc2 leads to metabolic re-initiation and proliferation, but at the expense of incurring substantial DNA damage. We suggest that eIF5A operates as a translatome remodeler that suppresses metabolism to shield the genome.

[1] Department of Biochemistry and Molecular Biology, Miller School of Medicine, University of Miami, Miami, FL 33136, USA. [2] Sylvester Comprehensive Cancer Center, Miller School of Medicine, University of Miami, Miami, FL 33136, USA. [3] Division of Hematology, Department of Medicine, Miller School of Medicine, University of Miami, Miami, FL 33136, USA. [4] The SickKids Proteomics, Analytics, Robotics & Chemical Biology Centre (SPARC Biocentre), The Hospital for Sick Children, Toronto, ON M5G 1×8, Canada. [5] Department of Urology, Miller School of Medicine, University of Miami, Miami, FL 33136, USA. [6] Present address: Bioinformatics Solutions Inc., Waterloo, ON N2L 6J2, Canada. [7] These authors contributed equally: Nathan C. Balukoff, J. J. David Ho. ✉email: stephenlee@med.miami.edu

The preservation of genomic integrity is essential for organismal survival and evolutionary fitness. This ability is especially critical during physiological stresses that damage DNA[1–4]. Anaerobic metabolism-induced extracellular acidosis (hereafter referred to as hypoxia or anaerobic acidosis) is one of the most frequently encountered stimuli/stresses in health and diseases, including development, exercise, cancer, and ischemia[5–9]. Aberrant proliferation, especially under nonpermissive growth environments, e.g., acidotic conditions, can lead to considerable DNA damage and chromosomal defects[10–13]. For these reasons, anaerobic acidosis induces metabolic depression to suppress energy consumption and inhibit growth[14–17]. This may explain why anaerobic acidosis exerts a protective effect during ischemic episodes in tumor microenvironments and various established cell lines or primary cultures[14–19].

As an emerging paradigm, the predominance of translation efficiency (TE) over mRNA-level fluctuations has been revealed as a primary mechanism of adaptation to physiological stimuli/stresses[20–24], as well as during evolution[25], development[26,27], differentiation[28], cell-type specificity[29], and circadian regulation[30], among others. Specifically, cells globally reprogram their protein-synthesis machinery in a stimuli-specific manner to produce unique, stress-adaptive translatomes (protein outputs). For instance, translation factors eIF3d[31], eIF4E2[21,24], eIF4G3[21], eIF4E3[22], eIF5B[20], and DAP5[32] have been demonstrated as critical components of specialized translation machineries in cells responding to various conditions. These translation factors operate as translatome remodelers that control protein outputs at the translational level (i.e., TE of mRNAs) to activate biochemical pathways. Given a model of translational plasticity and the drastic effects of anaerobic acidosis on cellular physiology, we hypothesized that cells activate a unique translational program to produce key proteins required to suppress metabolism and preserve genomic integrity. This hypothesis was tested using several system-level technologies, including our recently developed mass spectrometry analysis of translation factors using ribosome-density fractionation and isotopic labeling experiments[20] (MATRIX) platform, which generates snapshots of translation-factor distribution in free, monosomes, light and heavy polysome fractions under different cellular conditions.

In this study, we report the identification of a translation-efficiency gatekeeping mechanism that suppresses metabolism to prevent DNA damage in cells responding to anaerobic acidosis. This pathway requires the acidotic adaptive engagement of eIF5A, a protein that can be traced back to the last common universal ancestor (LUCA)[33,34], which evolved under anaerobic conditions[35]. We discuss a model by which eIF5A operates as a translatome remodeler that mediates metabolic depression to shield the genome.

## Results

**Anaerobic acidosis adaptively engages the eukaryotic translation factor 5A**. Physiological anaerobic acidosis was modeled by allowing hypoxic (1% $O_2$) cells to naturally acidify their extracellular milieu to pH 6.0, which recapitulates various in vivo conditions, e.g., ischemic tissues[36] and microenvironments of aggressive tumors[37,38]. Under these conditions, anaerobic acidosis induces an extreme phenotype characterized by a marked decrease in cellular ATP utilization (Supplementary Fig. 1a), transcription (Supplementary Fig. 1b), protein synthesis (Supplementary Fig. 1c), cellular proliferation (Supplementary Fig. 1d–f), and a prominent change in cell patterning (Supplementary Fig. 1g). Notably, cell viability is maintained under these conditions (Supplementary Fig. 1h, i). Anaerobic acidosis also activates the formation of physiological amyloid bodies (A

bodies)[16], which are involved in metabolic suppression and are used as an additional marker of adaptation to low pH (Supplementary Fig. 1j).

We hypothesized that cells confronted with anaerobic acidosis activate a unique translational program for suppressing metabolism and preserving genomic integrity. To test this hypothesis, we first performed an unbiased, high-throughput analysis of the translational architecture of cells adapting to hypoxia-induced acidosis using our recently developed MATRIX[20] platform, which discriminates translation factors based on their distribution in sucrose gradients (e.g., free, monosome, light, and heavy polysome fractions) (Fig. 1a, Supplementary Fig. 2a). MATRIX has an inherent bias toward translation-elongation factors associated with heavy polysomes, although initiation factors can also be readily detected in the heavy polysomal fractions, where they are primarily associated with initiating ribosomes on mRNAs already undergoing productive protein synthesis (i.e., those that contain multiple elongating ribosomes). Polysome-associated translation-initiation factors can be detected by mass spectrometry (MS) and western blot analysis under various experimental settings[20,24,39–45]. Metabolic pulse labeling with SILAC (pSILAC) allows us to identify and eliminate confounding signals from newly synthesized peptides. MATRIX successfully identified 51 canonical translation factors (each by at least 2 unique peptides). We applied a series of stringency criteria to identify the most promising candidate(s) (Supplementary Fig. 2b, MATRIX_sourcefile). We narrowed down on candidates (28 out of 50) that were detected across all fractions. Next, using the ratio of protein abundance in polysome fractions over ribosome-free fractions as our primary readout for translational engagement, we further narrowed our attention to those translation factors that exhibited at least a twofold increase in hypoxia acidosis (HA), in line with our previous studies[20] and a cutoff used by other groups[46,47], compared to basal conditions. Following outlier removal using the interquartile-range method, we eliminated candidates that were also found to be activated by hypoxia alone. Finally, we applied a twofold cutoff to our secondary readout, i.e., the ratio of protein abundance in polysome fractions over monosome (40/S/60S/80S) fractions, to eliminate candidates that may be stalled at the initiation step of translation and to improve our confidence that candidates are indeed associated with actively translating ribosomes. Translation factors that satisfied all the aforementioned criteria were pursued as the focus for further investigation.

Translation factors that are enriched in heavy polysomes of hypoxic acidotic cells compared to normoxic (Fig. 1b, Supplementary Fig. 2a) or hypoxic neutral cells (Fig. 1c, Supplementary Fig. 2a) are highlighted in dark-blue columns. Gray columns indicate translation factors whose distributions are not affected by anaerobic acidosis, while red (Fig. 1b, Supplementary Fig. 2a) and yellow (Fig. 1c, Supplementary Fig. 2a) columns represent translation factors that are relatively enriched in heavy polysomes under basal and hypoxia-neutral pH conditions, respectively. We selected candidate translation factors that displayed enrichment in heavy polysome/free of hypoxic acidotic compared to both normoxic neutral cells (Fig. 1b) and hypoxic neutral pH conditions (Fig. 1c). As mentioned above, we excluded translation factors that showed enrichment in polysome/free in both hypoxic/acidotic and hypoxic/neutral as these candidates were affected by [$O_2$] and not exclusively pH. This selection approach identified eIF5A as a translation factor that accumulates in heavy polysomes of hypoxic acidotic cells while excluding others, such as eIF4B, that display [$O_2$] sensitivity. Analysis using the ratio of heavy polysome/monosome protein abundance as a secondary readout further supported the relative enrichment of eIF5A in heavy polysomes under anaerobic acidosis compared to normoxic

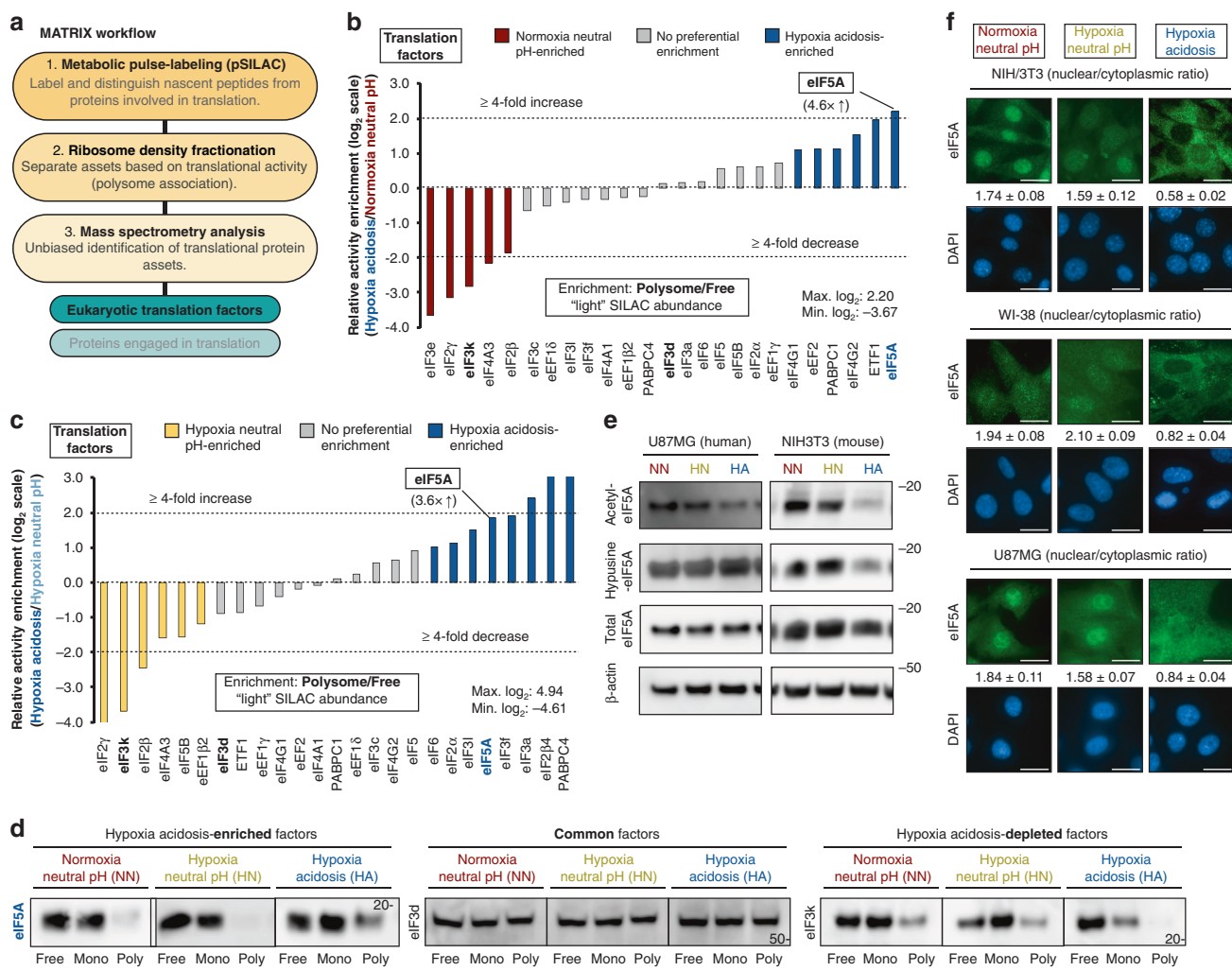

**Fig. 1 MATRIX analysis reveals eIF5A-adaptive engagement during anaerobic acidosis. a** Workflow schematic of MATRIX, an unbiased, high-throughput platform that measures the activity of translational assets. MATRIX analysis of differential translation-factor utilization in human cells (U87MG) exposed to hypoxia acidosis (1% $O_2$, pH 6.0, 24 h) compared to **b** basal (21% $O_2$, pH 7.4, 24 h) and **c** hypoxia-neutral pH (1% $O_2$, pH 7.4, 24 h) conditions, using the ratio of heavy polysome to free abundance as the readout. Hypoxia acidosis-activated translation factors (dark-blue bars), basal-activated translation factors (red bars), and hypoxia-neutral pH-activated translation factors (light-blue bars). **d** Representative immunoblots of U87MG ribosome-density fractions from indicated conditions. Mono: monosome fraction. Poly: polysome fractions. **f** Representative immunoblots of U87MG were subjected to indicated treatments. **e** Representative images of eIF5A immunocytochemistry in human (WI-38, U87MG) and mouse (NIH/3T3) cell lines subjected to indicated treatments. Data represent mean ± SEM ($n = 3$). Scale bars: 20 μm.

or hypoxic neutral conditions (Supplementary Fig. 2c, d). Reducing the stringency (to 1.5-fold compared to basal conditions, for example) resulted in additional candidates being included, e.g., eIF5B, eIF2S1, and eEF1G. However, neither of these candidates showed preferential activation in HA when compared to hypoxia alone. Consistent with datasets obtained with MATRIX and our selection approach, immunoblots of ribosome-density fractions showed an increase in heavy polysome-associated eIF5A in hypoxic acidotic cells compared to normoxic or hypoxic neutral conditions (Fig. 1d) that were not due to a change in its steady-state levels (Fig. 1e). The low pH-dependent decrease in eIF3k heavy-polysome association, and eIF3d that did not display any changes, was used as control (Fig. 1d). In support of a possible enhanced eIF5A translational activity in the cytoplasm, we observed a remarkable shift in steady-state eIF5A subcellular localization across species, from largely nuclear under normoxia- and hypoxia-neutral pH conditions to predominantly cytoplasmic during HA (Fig. 1f). Basal eIF5A nuclear localization has been attributed to the post-translational lysine acetylation[48,49]. Consistent with these

findings, we observed decreased eIF5A acetylation across species under HA compared to neutral pH conditions (Fig. 1e, Supplementary Fig 2e). These data support a positive association of eIF5A deacetylation with cytoplasmic activity and engagement with heavy polysomes, consistent with MATRIX analysis. Hypusination is a unique and stable eIF5A post-translational modification required for its activity[44,50–53]. eIF5A hypusination levels remained constant across all treatment conditions (Fig. 1e). Thus, we have identified eIF5A as a potential pH-regulated translation factor.

**eIF5A controls metabolic depression during anaerobic acidosis.** We tested the biological implications of adaptive engagement of eIF5A for metabolic adaptation during anaerobic acidosis. The results revealed that eIF5A knockdown led to a significant resumption of ATP utilization (Fig. 2a), transcription (Fig. 2b), and protein synthesis (Fig. 2c, Supplementary Fig 3c) in anaerobic acidotic cells but had only relatively modest effects in cells maintained in neutral conditions (Supplementary Fig. 3a, b). In

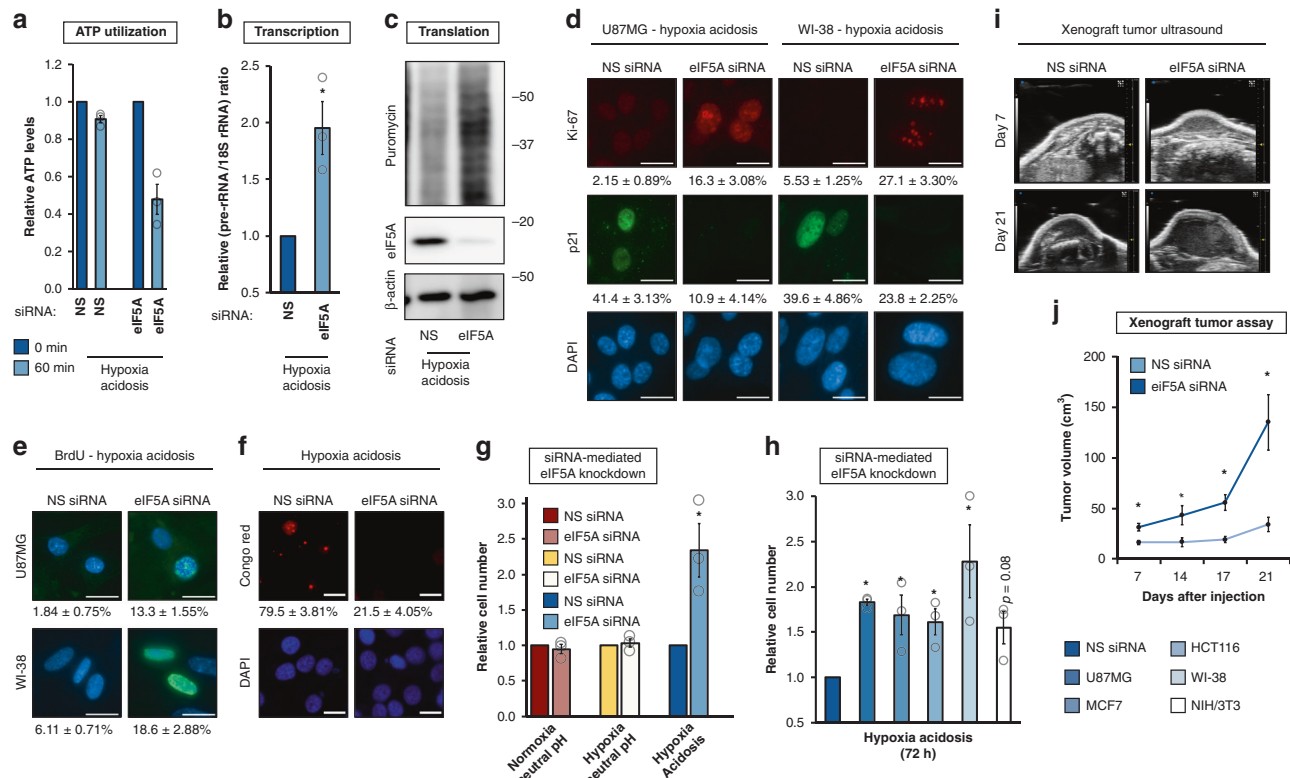

**Fig. 2 eIF5A enables metabolic depression during anaerobic acidosis. a** Relative ATP utilization, **b** transcriptional intensity, and **c** translational intensity in U87MG replete or depleted of eIF5A under hypoxia acidosis conditions. **a**, **b** NS nonsilencing. Data represent mean ± SEM ($n = 3$). An asterisk indicates $p = 0.024$, $0.028$ (**a**, **b**) compared to NS siRNA, two-sided student's $t$ test. **d** Ki-67 and p21 immunocytochemistry, **e** DNA replication (BrdU staining), and **f** Congo red staining for A bodies in U87MG replete or depleted of eIF5A under hypoxia acidosis conditions (72 h) ($n = 5$). Quantitation in (**d**) and (**e**) represents mean ± SEM ($n = 7$, 5). Scale bars: 20 μm. **g** Effect of eIF5A knockdown on steady-state cell numbers in U87MG subjected to indicated treatments. NS nonsilencing. Data represent mean ± SEM ($n = 3$). An asterisk indicates $p = 0.035$ compared to NS siRNA, two-sided student's $t$ test. **h** Effect of eIF5A knockdown on steady-state cell numbers in human and mouse cell lines under hypoxia acidosis conditions. NS nonsilencing. Data represent mean ± SEM ($n = 3$). An asterisk indicates $p = 0.001$, 0.001, 0.047, and 0.025 (U87MG, MCF7, HCT116, and WI-38) compared to the corresponding NS siRNA, two-sided student's $t$ test. **i** Representative ultrasound images and **j** tumor volume measurements of mouse xenograft tumor-formation assays using MCF7 replete or depleted of eIF5A and pretreated with hypoxia acidosis for 72 h. Data represent mean ± SEM ($n = 6$). An asterisk indicates $p = 0.021$, 0.031, 0.004, and 0.014 (days 7, 14, 17, and 21) compared to NS siRNA, two-sided student's $t$ test.

addition, eIF5A silencing led to an increase in cellular proliferation as demonstrated by Ki-67/p21 staining (Fig. 2d), BrdU labeling (Fig. 2e), and flow cytometry analysis (Supplementary Fig. 3d). We also observed a decreased number of A bodies (Fig. 2f) and a reversal of cellular patterning (Supplementary Fig. 3e). Overall, these effects of eIF5A knockdown resulted in increased cell number under anaerobic acidosis, but not basal or hypoxia-neutral pH conditions (Fig. 2g). Silencing eIF5A in hypoxic/acidotic conditions did not completely restore transcription/translation activity to levels seen in neutral cells but was sufficient to sustain cellular proliferation and viability (Supplementary Fig. 3a, b). The effects of eIF5A knockdown in HA were observed across a variety of human and mouse cell types, including normal human fibroblast WI-38 (Fig. 2h). Expanding these observations in vivo using mouse xenograft assays, eIF5A silencing in cancer cells pretreated with HA led to a significant increase in tumor growth (Fig. 2i, j, Supplementary Fig. 3f). Hypoxia and hypoxia-induced acidosis are hallmarks of the tumor microenvironment that contribute to drug resistance[54], an observation that we reproduced (Supplementary Fig. 3h). Notably, eIF5A depletion during HA significantly increased the sensitivity of cancer cells to conventional antiproliferative drugs, e.g., vincristine and temozolomide (Supplementary Fig. 3h). Consistent with published reports[55,56], treatment with the global hypusination inhibitor GC7[57] prevented proliferation under basal

conditions and did not lead to an increase in cell number in HA as we observed with specific eIF5A silencing (Supplementary Fig. 3i). These findings precluded its use as a surrogate for eIF5A-specific silencing. Overall, these data suggest that eIF5A suppresses metabolism in cells exposed to anaerobic acidosis.

**A screen by TMT-pSILAC identifies pH-regulated proteins.** Global remodeling of the translational landscape in response to physiological stimuli involves changes in both translation machinery and protein output. To understand the mechanisms by which eIF5A controls metabolic depression during HA, we first performed a tandem mass tag-pulse SILAC (TMT-pSILAC) and MS screen on cells subjected to basal, hypoxia-neutral pH, and HA (Fig. 3a, b, Supplementary Fig. 4a, b, TMT-pSILAC_source-file). This approach allows us to compare newly made proteins under each condition in addition to changes in steady-state protein levels (Supplementary Fig. 4a). The TMT-pSILAC screen identified 244 proteins whose productions are enhanced under anaerobic acidosis conditions compared to normoxic/neutral and hypoxic/neutral conditions (Fig. 3c). Here, we chose 1.5-fold as the threshold for enhancement based on the average induction of glycolytic enzymes in hypoxia-neutral compared to normoxia-neutral[58–60] (Supplementary Fig. 4b, TMT-pSILAC_sourcefile). Well-characterized candidates uncovered by TMT-pSILAC with available high-quality reagents and that displayed various fold

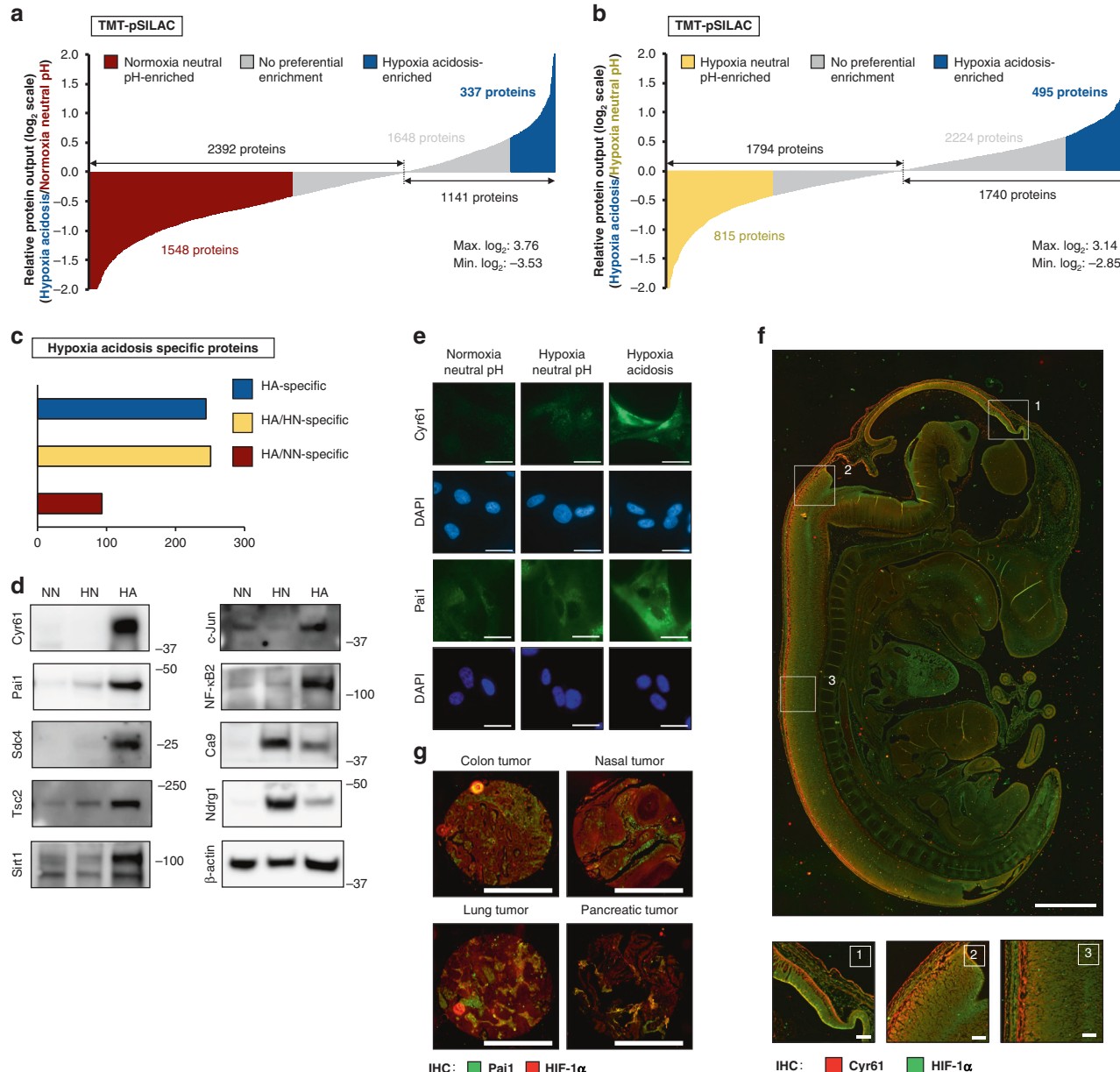

**Fig. 3 TMT-pSILAC identifies the translatome of anaerobic acidosis-dependent metabolic depression.** Proteomic output analysis by TMT-pSILAC followed by mass spectrometry cells subjected to hypoxia acidosis (HA) versus **a** basal normoxia-neutral pH (NN) and **b** hypoxia-neutral pH (HN) conditions. HA enrichment: ≥1.5× increase; NN/HN enrichment: ≤0.75× decrease. **c** Graph depicting HA-enriched proteins (≥1.5× increase), blue = proteins that are upregulated in HA compared to both conditions, yellow proteins that are upregulated in HA compared to HN but not NN, red = proteins that are upregulated in HA compared to NN but not HA. Representative **d** immunoblots and **e** immunocytochemistry images of hypoxia-acidosis (HA)-specific translatome targets in U87MG. NN normoxia-neutral pH, HN hypoxia-neutral pH, n = 5. Scale bars: 20 μm. **f** Representative images of Cyr61 (HA-specific marker, red) and HIF-1α (classic hypoxia marker, green) immunohistochemistry in embryonic day 12 (ED12) CD1 mouse embryo. Scale bars: 1000 μm. Enlarged images of indicated sections (top) are presented on bottom panels (n = 3). Scale bars: 100 μm. **g** Representative images of Pai1 (HA-specific marker, green) and HIF-1α (hypoxia marker, red) immunohistochemistry in human hypoxic tumor core sections (n = 3). Scale bars: 1000 μm.

enrichment above a threshold of 1.5-fold[61,62] were tested by immunoblot analysis (Fig. 3d, Supplementary Fig. 4c). Several candidates were also confirmed by mRNA TE analysis (Supplementary Fig. 4d) and immunocytochemistry (Fig. 3e). We define TE as the ratio of mRNA abundance in polysome fractions to that in free/monosome fractions. We note the difference between this definition from its usage in other studies, e.g., ribosome-profiling experiments[63]. These acidosis-enriched proteins represent more reliable and specific biomarkers of low extracellular pH, in contrast to hypoxia-inducible proteins, e.g., Ca9, which are often used as surrogate markers of acidosis[64,65] (Fig. 3d). Additional classic

hypoxia-inducible proteins (e.g., Ca9 and Ndrg1) that are induced in both hypoxia-neutral pH and acidotic conditions were measured as controls to demonstrate the HA specificity of identified targets (Fig. 3d). Next, we tested the potential utility of these proteins as in vivo markers of acidosis, using immunohistochemistry on mouse embryos (Fig. 3f, Supplementary Fig. 4e) and hypoxic tumor core sections (Fig. 3g, Supplementary Fig. 4f). We note the prominent Cyr61 signal that exhibits a gradual overlap with HIF-1α (classic hypoxia marker) in the developing mouse central nervous system, which requires hypoxia signaling for its proper development[66] (Fig. 3f). Areas of overlap between

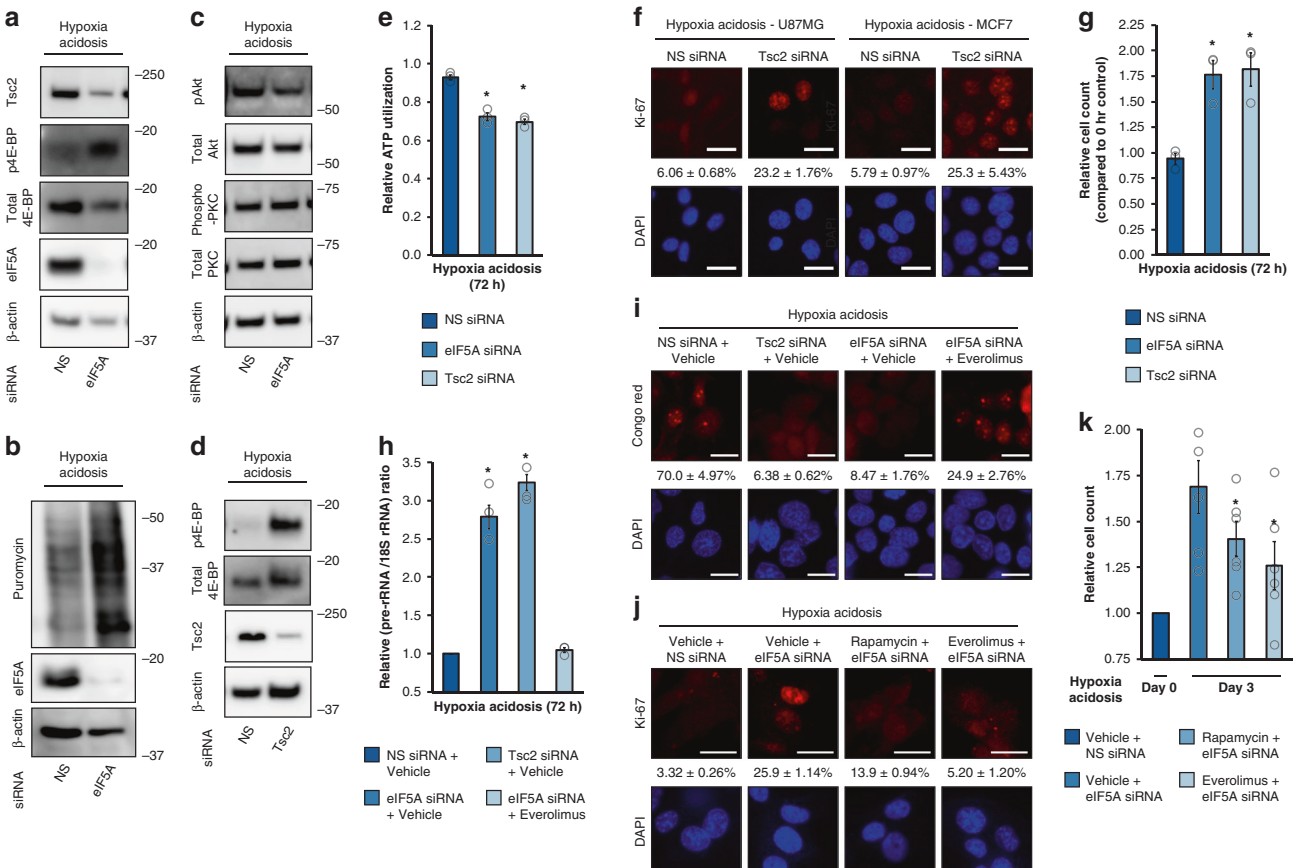

**Fig. 4 eIF5A suppresses global metabolism and proliferation through Tsc2/mTORC1. a–c** Representative immunoblots of U87MG replete or depleted of eIF5A under hypoxia-acidosis conditions ($n = 3$). **d** Representative immunoblots of U87MG replete or depleted of Tsc2 under hypoxia-acidosis conditions ($n = 3$). **e** Effects of eIF5A and Tsc2 knockdown on cellular ATP levels under hypoxia-acidosis conditions. NS nonsilencing. Data represent mean ± SEM ($n = 3$). An asterisk indicates $p = 0.024$ and $0.012$ (eIF5A siRNA and Tsc2 siRNA) compared to NS siRNA, two-sided student's $t$ test. **f** Representative images of Ki-67 immunocytochemistry in U87MG (left panel) and MCF7 (right panel) replete or depleted of Tsc2 under hypoxia-acidosis conditions. NS nonsilencing. Data represent mean ± SEM ($n = 5$). Scale bars: 20 μm. Effect of Tsc2 knockdown on U87MG **g** cell number, **h** transcriptional intensity, and **i** Congo red staining for A bodies. Scale bars: 20 μm. NS nonsilencing. Data represent mean ± SEM ($n = 3$ (**g**, **h**); $n = 5$ (**i**)). An asterisk indicates $p = 0.046$ and $0.042$, $p = 0.007$ and $0.002$ (**g**, **h** eIF5A siRNA and Tsc2 siRNA) compared to NS siRNA, two-sided student's $t$ test. **j** Representative images of Ki-67 immunocytochemistry in U87MG replete or depleted of eIF5A and treated with mTORC1 inhibitors rapamycin and everolimus under hypoxia-acidosis conditions. NS nonsilencing. Data represent mean ± SEM ($n = 5$). Scale bars: 20 μm. **k** Effect of mTORC1 inhibition on eIF5A-replete or -depleted U87MG cell number under hypoxia-acidosis conditions. NS nonsilencing. Data represent mean ± SEM ($n = 6$). An asterisk indicates $p = 0.026$ and $0.022$ (Rapamycin + eIF5A siRNA, Everolimus + eIF5A siRNA) compared to Vehicle + eIF5A siRNA, two-sided student's $t$ test.

acidosis markers and HIF-1α were also observed in tumor cores (Fig. 3g). Together, this screen revealed potential effectors of physiological anaerobic acidosis.

**eIF5A regulates the mTORC1 inhibitor Tsc2 to suppress the metabolism.** TMT-pSILAC analysis and various assays (Fig. 3d, Supplementary Fig. 4c, d) revealed tuberous sclerosis complex 2 (Tsc2) as a prominent HA-specific protein. Tsc2[67,68] is a major activity repressor of the mechanistic target of rapamycin (mTOR) complex 1 (mTORC1), a master growth regulator that integrates environmental signals to orchestrate responsive changes in cellular proliferation, metabolism, and especially protein synthesis[69–72]. Hypophosphorylation of a key mTORC1 target eIF4E-binding protein (4E-BP) results in the potent inhibition of protein synthesis and suppression of metabolism[72,73]. Interestingly, Tsc2-mediated suppression of mTOR under acidotic conditions has been reported[74]. Here, we found that eIF5A controls global translation by mediating HA-specific Tsc2 induction. During HA, eIF5A knockdown by pooled or single-site small-interfering RNA

(siRNA) attenuates Tsc2 protein induction and derepresses mTORC1 activity, resulting in increased 4E-BP phosphorylation and enhanced global translational intensity (Fig. 4a, b, Supplementary Fig. 5a–c, Fig. 2c). In contrast to mTORC1, mTORC2 activity[75–77] was largely unaffected by eIF5A depletion during HA (Fig. 4c, Supplementary Fig. 5d). We found that AMPK had no effect on steady-state Tsc2 phosphorylation in acidosis, consistent with previous reports that acidosis inhibits AMPK activity[78,79] (Supplementary Fig. 5e).

Silencing of Tsc2 was sufficient to phenocopy the effects of eIF5A depletion during anaerobic acidosis in terms of increasing 4E-BP phosphorylation, reactivating mTORC1 activity (Fig. 4d, Supplementary Fig. 5f), increased ATP utilization (Fig. 4e), proliferative signals (Fig. 4f), cell number (Fig. 4g), RNA synthesis (Fig. 4h), and A-body formation (Fig. 4i). Furthermore, drug-mediated inhibition of mTOR activity during anaerobic acidosis significantly attenuated the effects of eIF5A depletion on RNA synthesis (Fig. 4k), A-body formation (Fig. 4i), proliferation (Fig. 4j), and cell number (Fig. 4k). Thus, the eIF5A/Tsc2/mTORC1 axis suppresses metabolism during anaerobic acidosis.

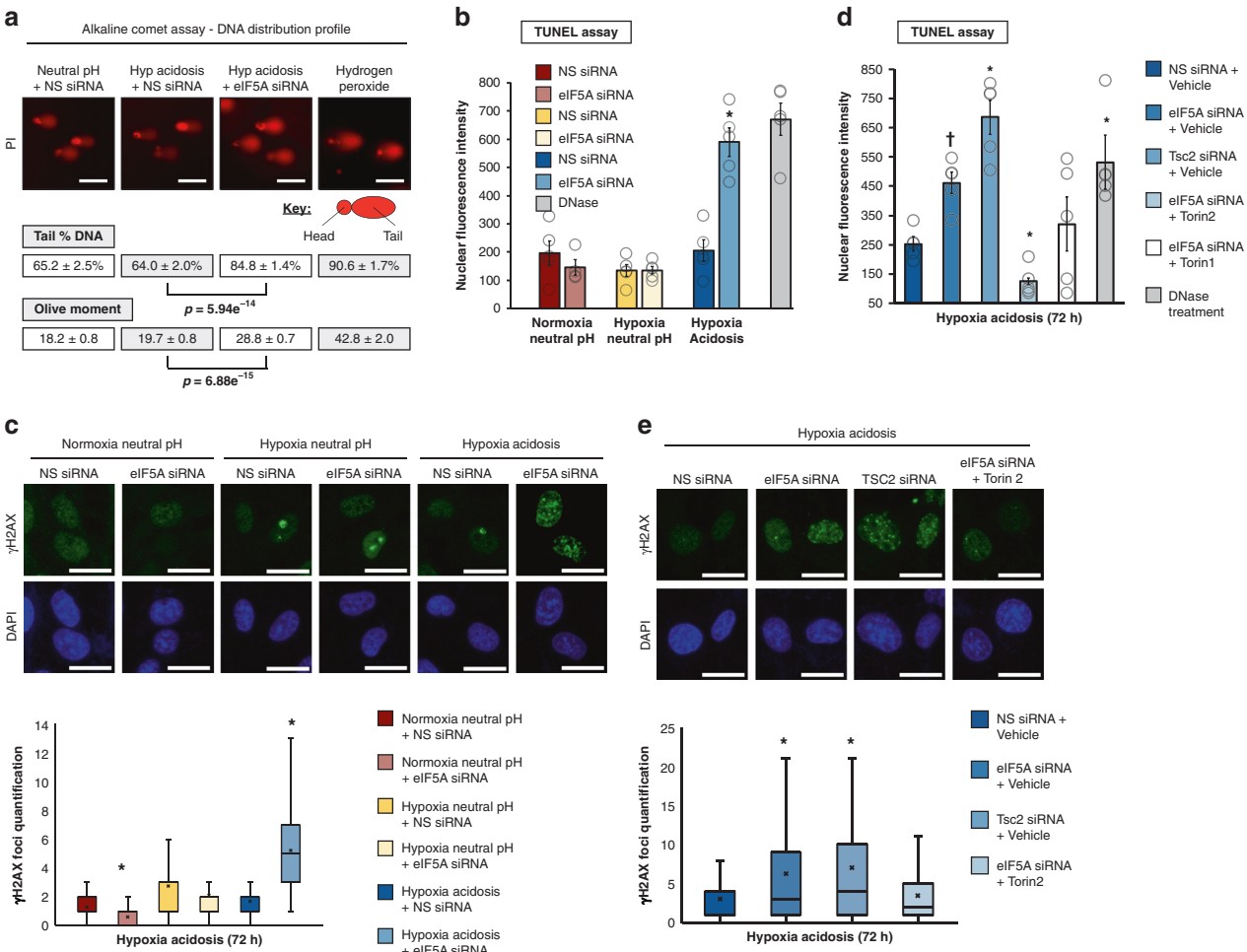

**Fig. 5 The eIF5A/Tsc2 axis averts DNA damage.** Assessment of the effect of eIF5A silencing on DNA damage by **a** alkaline comet analysis ($n = 3$) and **b** TUNEL measurements in U87MG subjected to indicated conditions. Data represent mean ± SEM ($n = 5$). Asterisk indicates $p = 0.001$ compared to NS siRNA, two-sided student's $t$ test. **c** (Top) Representative images of γH2AX foci in U87MG subjected to indicated conditions. Scale bars: 20 µm. (Bottom) analysis of γH2AX foci ($n = 5$), asterisk indicates $p = 0.00001$ and $0.00001$ (NN eiF5A siRNA, HA eIF5A siRNA) compared to NS siRNA, two-sided Mann–Whitney $U$ test. The top of the box denotes Q3, the bottom of the box represents Q1; middle line denotes median; X represents mean; bottom whisker denotes minimum: 1st quartile—(1.5*IQR); top whisker denotes maximum: 3rd quartile + (1.5*IQR). **d** TUNEL analyses of the effects of Tsc2 knockdown and mTORC1 inhibition (by Torin 1 and 2) on DNA damage in cells replete or depleted of eIF5A under hypoxia-acidosis conditions. Data represent mean ± SEM ($n = 5$). An asterisk indicates $p = 0.005$, $0.004$, $0.002$, and $0.027$ (eIF5A siRNA + vehicle, Tsc2 siRNA + vehicle, and eIF5A siRNA + torin2) compared to NS siRNA + Vehicle. † indicates $p = 0.006$ eIF5A siRNA compared to Vehicle and Tsc2 siRNA + Vehicle, two-sided student's $t$ test. **e** Top panel: representative images of γH2AX foci in U87MG cells depleted of Tsc2, and in eIF5A-replete or -depleted cells treated with mTORC1 inhibitors (Torin 1 and 2) under hypoxia-acidosis conditions. Bottom panel: analysis of γH2AX ($n = 5$), an asterix indicates $p = 0.0005$ and $0.00001$ (eIF5A siRNA, tsc2 siRNA) compared to NS siRNA + vehicle, two-sided Mann–Whitney $U$ test. Top of the box denotes Q3, the bottom of the box represents Q1; middle line denotes median; X represents mean; bottom whisker denotes minimum: 1st quartile—(1.5*IQR); top whisker denotes maximum: 3rd quartile + (1.5*IQR).

**The eIF5A/Tsc2/mTORC1 axis prevents DNA damage**. It is ostensibly curious that eIF5A, one of the most highly conserved proteins, would prevent cellular growth (Fig. 2). We reasoned that perhaps eIF5A functions as a proliferative brake to safeguard genome integrity during anaerobic acidosis, which significantly increases DNA and chromosomal damage if proliferation is unchecked under such conditions[1–4]. Indeed, eIF5A-competent cells effectively maintained genomic integrity during acidosis, as determined by DNA-damage measurements using alkaline comet (Fig. 5a) and terminal deoxynucleotidyl transferase dUTP nick end labeling (TUNEL) (Fig. 5b) assays, as well as γH2AX foci formation and levels[80] (Fig. 5c, Supplementary Fig. 6a). In contrast, eIF5A knockdown led to a significant increase in DNA damage, providing evidence for eIF5A as a preserver of genome stability during anaerobic acidosis (Fig. 5a–c, Supplementary

Fig. 6a). Next, we confirmed the role of the eIF5A/Tsc2/mTORC1 axis in protecting genome integrity. Tsc2 knockdown significantly increased DNA damage in a manner similar to eIF5A (Fig. 5d, e, Supplementary Fig. 6b), while mTOR inhibition effectively attenuated eIF5A silencing-induced DNA damage (Fig. 5d, e, Supplementary Fig. 6b). Taken together, these results suggest that adaptive engagement of eIF5A prevents DNA damage through Tsc2 protein induction to suppress mTORC1 activity and cellular proliferation.

**Nuclear export of Tsc2 mRNA by eIF5A increases its TE**. Next, we examined the mechanisms by which eIF5A controls Tsc2 protein induction during anaerobic acidosis. RNA immunoprecipitation experiments revealed that eIF5A specifically associated with the mRNAs of Tsc2 and two other eIF5A-regulated proteins

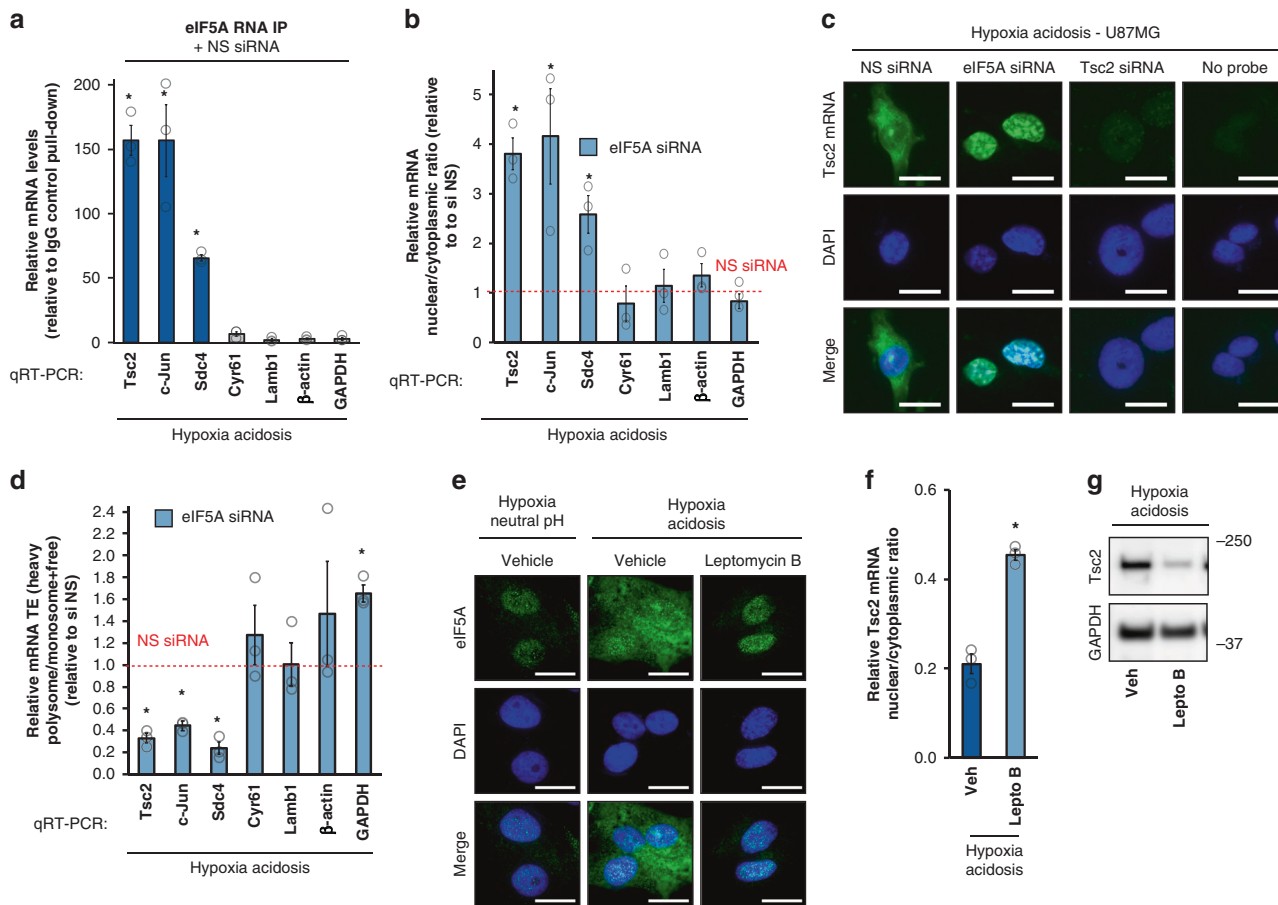

**Fig. 6 Mechanism of eIF5A-mediated Tsc2 protein induction. a** Co-immunoprecipitated mRNA levels of eIF5A-regulated and nonregulated mRNAs relative to IgG isotype control pulldown. Data represent mean ± SEM ($n = 3$). An asterisk indicates $p = 0.005$, 0.031, and 0.001 (Tsc2, c-Jun, and Sdc4) compared to IgG control, two-sided student's $t$ test. **b** Effect of eIF5A knockdown on mRNA subcellular localization of indicated mRNAs under hypoxia acidosis conditions. NS nonsilencing. Data represent mean ± SEM ($n = 3$). An asterisk indicates $p = 0.007$, 0.041, and 0.027 (Tsc2, c-Jun, and Sdc4) eIF5A siRNA compared to NS siRNA, two-sided student's $t$ test. **c** mRNA fluorescent in situ hybridization. NS nonsilencing. One-third exposure level was used for Tsc2 mRNA FISH under eIF5A siRNA conditions relative to all other conditions to avoid signal saturation ($n = 5$). Scale bars: 20 µm. **d** Effect of eIF5A knockdown on translation efficiencies of indicated mRNAs under hypoxia-acidosis conditions. NS nonsilencing. Data represent mean ± SEM ($n = 3$). An asterisk indicates $p = 0.0004$, 0.002, and 0.006 (Tsc2, c-Jun, and Sdc4) eIF5A siRNA compared to NS siRNA, two-sided student's $t$ test. **e** Effect of leptomycin B treatment on eIF5A protein subcellular localization in U87MG under indicated conditions. Vehicle: DMSO ($n = 5$). Scale bars: 20 µm. **f** Effect of leptomycin B treatment on Tsc2 mRNA subcellular localization under hypoxia-acidosis conditions. Vehicle (Veh): DMSO. Data represent mean ± SEM ($n = 5$). An asterisk indicates $p = 0.015$ compared to DMSO vehicle, two-sided student's $t$ test. **g** Representative immunoblot of Tsc2 protein levels in U87MG treated with leptomycin B ($n = 3$).

c-Jun and Scd4 but not the eIF5A-independent Cyr61, or control transcripts (Fig. 6a, Supplementary Fig. 7a, b). Consistent with these results, nuclear export of Tsc2, c-Jun, and Scd4 (Fig. 6b, c, Supplementary Fig. 7c) mRNAs during anaerobic acidosis was dependent on eIF5A, resulting in increased engagement by cytoplasmic polysomes (Fig. 6d) without significantly affecting steady-state mRNA expression (Supplementary Fig 7d). In stark contrast, eIF5A depletion had no effect on nuclear export and polysome engagement of eIF5A-independent mRNAs (Fig. 6b, Fig. 6d, Supplementary Fig. 7d) or on the levels of transporters CRM-1[81–83] and Xpo4[84] involved in eIF5A nuclear export (Supplementary Fig. 7e). These results provided evidence that eIF5A specifically interacts with its regulated mRNAs, but not other transcripts, to control their nuclear export and TE. In agreement with this, treatment with leptomycin B, which inhibits eIF5A nuclear export[83,85] (Fig. 6e), prevented anaerobic acidosis-induced cytoplasmic export of Tsc2 mRNA (Fig. 6f), leading to a reduction in steady-state Tsc2 protein levels (Fig. 6g). eIF5A has been shown to promote efficient translation elongation mainly

through polyproline motifs[86,87]. The depletion of eIF5A under anaerobic acidotic conditions did not affect steady-state protein and mRNA levels of ectopically expressed FLAG-tagged Tsc2 open-reading frame-only constructs (Supplementary Fig. 7f, g). Likewise, substitutions of two canonical triple-proline motifs at amino acid positions 540–542 and 1262–1264 did not affect ectopic Tsc2 protein and mRNA expression under HA conditions, regardless of eIF5A-knockdown status (Supplementary Fig. 7f, g). Taken together, these results suggest that eIF5A regulates target-TE during anaerobic acidosis primarily through mRNA export from the nucleus to cytoplasmic translating ribosomes.

**pH-sensing Sirt1 modulates eIF5A during fermentation.** Finally, we examined the mechanisms by which eIF5A is engaged during anaerobic acidosis. eIF5A shifts from nuclear in neutral pH to cytoplasmic during HA, a process that is associated with its deacetylation[48,49] (Fig. 7a, b, Supplementary Fig 8a, b, Fig. 1e, f).

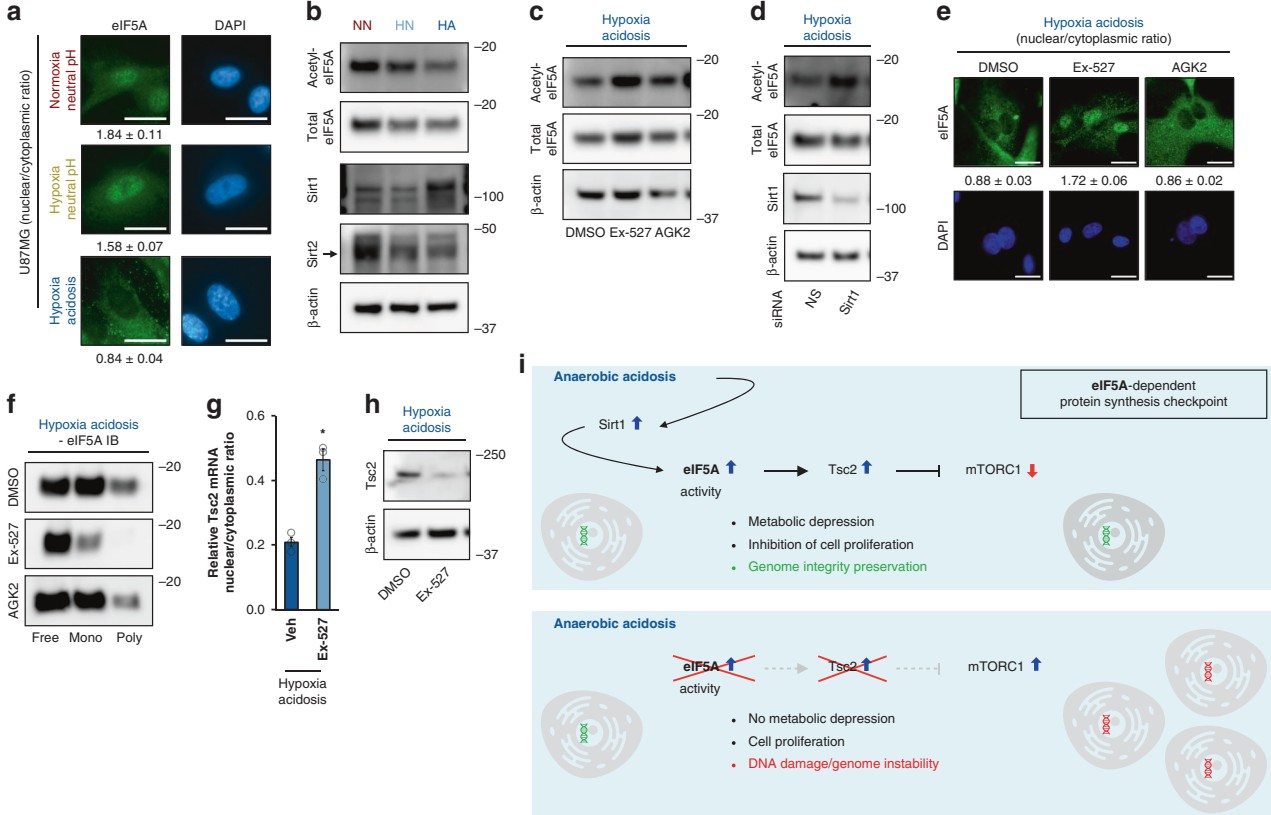

**Fig. 7 pH-sensing Sirt1 modulates eIF5A through deacetylation. a** Representative images of eIF5A immunocytochemistry showing eIF5A subcellular localization in U87MG subjected to indicated conditions. Data represent mean ± SEM ($n = 5$). Scale bars: 20 μm. **b** Representative immunoblots of U87MG subjected to indicated conditions. NN normoxia-neutral pH, HN hypoxia neutral, HA hypoxia acidosis ($n = 3$). Representative **c** immunoblots ($n = 3$) and **e** immunocytochemistry images ($n = 5$) of U87MG were treated with the indicated compounds under the indicated conditions. Scale bars: 20 μm. Ex-527: Sirt1 inhibitor; AGK2: Sirt2 inhibitor; DMSO: vehicle. **d** Representative immunoblots of U87MG depleted of Sirt1 ($n = 3$). **f** Representative immunoblots of U87MG ribosome-density fractions treated with the indicated compounds. Mono: light monosome fraction. Poly: heavy-polysome fractions ($n = 3$). **g** Effect of Sirt1 inhibition (using ex-527) on Tsc2 mRNA subcellular localization under hypoxia-acidosis conditions. NS nonsilencing. Data represent mean ± SEM ($n = 3$). An asterisk indicates $p < 0.05$ compared with DMSO vehicle, two-sided student's $t$ test. **h** Representative immunoblots of U87MG treated with the Sirt1 inhibitor ex-527 ($n = 3$). **i** Summary model of the Sirt1/eIF5A/Tsc2/mTORC1 pathway that enables metabolic depression and proliferative inhibition during anaerobic acidosis to prevent DNA damage.

Interestingly, the TMT-pSILAC screen uncovered Sirt1 protein induction by anaerobic acidosis (Fig. 3d, Fig. 7b, Supplementary Fig. 8c). Sirt1 and Sirt2 are two major eIF5A deacetylases in cells and in enzyme/substrate in vitro assays[48] that are regulated by $NAD^+$, a principal product of fermentation[88,89]. We found that Sirt1 and Sirt2 specifically deacetylate eIF5A during anaerobic acidosis and in neutral conditions, respectively (Fig. 7c, d, Supplementary Fig. 8e). Likewise, Sirt1, but not Sirt2 inhibition, prevented eIF5A cytoplasmic localization (Fig. 7e), reduced eIF5A association with translating ribosomes (Fig. 7f), and attenuated Tsc2 mRNA cytoplasmic export (Fig. 7g) and protein induction (Fig. 7h) under HA conditions. These results suggest that Sirt1 operates as a sensor of extracellular pH that adaptively upregulates eIF5A engagement during physiological anaerobic acidosis.

## Discussion

Here, we provide evidence that eIF5A suppresses metabolism to preemptively shield cells from DNA damage. This preventive program operates upstream and likely synergizes with reactive mechanisms induced in response to DNA damage, e.g., p53[90]. We propose the following model (Fig. 7i): Sirt1-mediated deacetylation adaptively modulates nuclear eIF5A in cells responding to anaerobic acidosis. eIF5A transports nuclear Tsc2 mRNA, among other targets, to cytoplasmic ribosomes for efficient translation during acidosis. Tsc2 inhibits mTORC1 and metabolism, enabling cells to escape DNA damage during anaerobic acidosis. Our findings emphasize the emerging concept of translatome remodelers that control critical biochemical pathways through protein output reprogramming and highlight the nascent paradigm of system-wide translational remodeling as a primary mechanism of biological adaptation. The demonstration that the Sirt1/eIF5A/Tsc2/mTORC1 axis induces metabolic depression may provide a mechanistic explanation for the longstanding observation that extracellular acidosis exerts protective effects during ischemic episodes, in tumor microenvironments, and various cell lines or primary cultures[14–17]. This axis may be broadly applicable to other physiological conditions and/or compound treatments are known to activate Sirt1[91,92], suppress mTORC1[93,94], and may potentially involve eIF5A2 in specific settings.

Initially identified as a translation-initiation factor[95,96], eIF5A also participates in translation elongation[44,97,98], termination[87,97], mRNA stability[99,100], and mRNA nuclear/cytoplasmic transport[82,84,101,102]. We show here that anaerobic acidosis is a potent engager of eIF5A, triggering remarkable changes in its subcellular localization from the nucleus to the cytoplasm, increased engagement with cytoplasmic heavy polysomes, and augmented translational activity compared to basal conditions.

Our data suggest that a principal role of eIF5A in acidotic cells is to export target mRNAs from the nucleus to the cytoplasm for efficient translation, a function of eIF5A that has been observed by other groups in cells responding to stimuli[82,101,102]. Our data do not exclude that eIF5A may be involved in the translation of polyproline motifs of other proteins under acidotic conditions[86]. Indeed, it will be interesting to compare the functions of adaptively engaged cytoplasmic eIF5A in acidosis with the roles of eIF5A under basal conditions (e.g., translation elongation). The data shown here also provide a physiological context and a potential mechanism for the protective role of eIF5A knockdown in anoxia[103] and the hypothesized tumor-suppressor role of eIF5A[104], although we are cognizant that this may depend on the genetic background of various cellular models[55,56,105,106]. The metabolic depression observed in response to HA resembles what has been described as cellular dormancy in the literature[107–109]. Engagement of eIF5A by the acidic tumor microenvironment may confer the dormant phenotype to tumor cells that is believed to increase resistance to traditional therapies and promote metastasis long after the removal of the primary lesion. We did notice a significant increase in tumor size and sensitivity of tumor cells to conventional antiproliferative drugs following depletion of eIF5A. The efficacy of cancer treatments could potentially be improved by targeting eIF5A alongside routine chemotherapies. On the other hand, pharmacological activation of the Sirt1/eIF5A/Tsc2/mTOR axis and ensuing metabolic depression may help cells sustain viability while maintaining genome integrity during ischemic episodes. We point out that even though the current study is focused on eIF5A, our unbiased MATRIX screen revealed additional candidate proteins that may be activated by HA, including some that are not known to participate in protein synthesis. In addition, the important loss of polysomes observed during hypoxia/acidosis likely as a consequence of Tsc2-mediated mTOR inhibition may also cause a relative enrichment of polysomic eIF5A, among other translation factors, detected by MATRIX. Further studies are warranted to investigate the role of additional HA-activated translation factors and regulators.

Interestingly, eIF5A is among a handful of eukaryotic genes that can be traced back to the LUCA[33], which evolved under anoxic conditions and relied on anaerobic metabolism[35]. Thus, it is plausible that eIF5A was retained in modern eukaryotes for adaptation to anaerobiosis, in addition to its basal activities[51,55]. It is ironic that cells activate a dedicated translational program under conditions that have been historically perceived as inhibitory for global protein synthesis. The identification of an acidosis-enriched translatome and HA-engaged translation factors, e.g., eIF5A, highlights the evolving concept of global translational reprogramming as a central mechanism of adaptation. The data also provide another layer of support to the concept that protein concentrations evolve under stricter pressure than mRNA levels[25]. The future challenge will be to integrate the concept of translatome remodelers in fields that have been traditionally dominated by the study of the transcriptional response to stimuli.

## Methods

**Cell culture and reagents**. Human and mouse cell lines used in this study, i.e., U87MG, MCF7, PC3, A549, HCT116, WI-38, and NIH/3T3 were purchased from the American Type Culture Collection (ATCC) and propagated in Dulbecco's modified Eagle's medium (DMEM) (HyClone) with 10% fetal bovine serum (FBS) (Omega Scientific) and 1% penicillin–streptomycin (HyClone). Cells were maintained at 37 °C in a 5% $CO_2$-humidified incubator (NN). Cells were subjected to hypoxia (HN) (1% $O_2$, 24 h, unless otherwise stated) at 37 °C in a 5% $CO_2$, $N_2$-balanced, humidified H35 HypOxystation (HypOxygen). Cells were subjected to HA by culturing them in acidotic permissive media in the hypoxic chamber. Cells were introduced to DMEM without bicarbonate at pH 7.4. They were allowed to naturally acidify to around a pH of 6.0 (approximately 30 min). Ex-527 (VWR, SIRT1-specific inhibitor) was used at a final concentration of 4 μM, and AGK2 (VWR, Sirt2-specific inhibitor) was used at a final concentration of 5 μM; cells were

pretreated with the drug for 24 h before being placed in their respective conditions for 24 h (normoxia-neutral pH, hypoxia-neutral pH, and HA). Rapamycin (Sigma) was used at a final concentration of 200 nM; cells were pretreated with the drug for 1 h before being placed in their respective condition. Torin 1 and Torin 2 were used at a final concentration of 250 nM; cells were pretreated with the drug for 1 h before being placed in their respective condition. Leptomycin B (BioVision) was used at a final concentration of 50 ng/μL; cells were pretreated with the drug for 1 h before being placed in their respective condition. GC7 (EMD Millipore) was used at a concentration of 25 μM; cells were pretreated for 24 h before being placed in their respective condition.

**Immunocytochemistry**. Cells were fixed on coverslips for 10 min using 4% formaldehyde (in phosphate-buffered saline (PBS)), washed with PBS, and then permeabilized for 10 min using 0.5% triton (in PBS). After washing, cells were blocked with 5% FBS for 1 h. Cells were incubated for 1 h at 37 °C in primary antibody (1:100), washed, and incubated with the corresponding secondary antibody (1:500) for 1 h at 37 °C. Cells were washed three times for 10 min, and nuclei were stained using Hoescht 3358 (1:1000, Thermo) during the second wash. Cells were mounted on slides using Flouromount and visualized using fluorescent microscopy. Antibodies used: KI-67 (SantaCruz, sc-23800), P21 (Proteintech, 10355-1-AP), EIF5A (abcam, ab32443), acetylated EIF5A (Lys47) (Boster Bio, P01727), CYR61 (Proteintech, 26689-1-AP), and PAI1 (Proteintech, 13801-1-AP). For ki-67 and p21 analysis, the number of nuclei with positive staining was compared to the total number of nuclei. To determine nuclear/cytoplasmic ratio, the corrected total cell fluorescence was determined for both the nucleus and cytoplasm using ImageJ [CTCF = integrated density − (area of selected cell × mean fluorescence of background readings)] and compared = $CTCF^{nucleus}/CTCF^{cytoplasm}$.

**DNA replication/BrdU assay**. BrdU pulse experiments were done with Roche Brdu Labeling and Detection Kit 1 (Roche, 11 296 736 001) according to the manufacturer's protocol. Briefly, cells were pulsed for 30 min with 1:1000 BrdU, fixed with Ethanol Fixative, incubated with Anti-BrdU working solution (1:10), and incubated with Anti-mouse-Ig-fluorescein (1:10); nuclei stained with Hoescht 3358 (1:1000, Thermo) and mounted using fluormount. The number of nuclei with BrdU labeling was counted and compared to the total number of nuclei.

**Propidium iodide (PI) and fluorescein diacetate (FDA) staining**. Live cells were incubated with a final concentration of 1 μg/ml each of PI, FDA, and Hoescht 3342 (ThermoFisher Scientific), for 30 min at 37 °C and washed twice with media before imaging by fluorescence microscopy.

**Cell counting**. Equal number of cells were plated on individual plates, and a Day-0 count (when cells were switched to their respective conditions, e.g., NN, HN, and HA) was performed using a hemocytometer. Cells were allowed to incubate for three days in conditions and counted on Day 3. Cell number was compared between Day 0 and Day 3. For cell death analysis, Day 3 cells were incubated in 1:1 solution of Trypan Blue and the number of blue cells (dead) was counted and compared to the total number of cells.

**Congo red staining**. Cells were fixed with 4% formaldehyde for 10 min and then permeabilized with 0.5% triton for 10 min. Cells were washed with PBS and stained with Hoescht58 (1/1000) for 10 min. Cells were washed with $ddH_2O$ and then incubated with Congo Red solution (3.5 mM Congo Red, 0.5 M NaCl, and 80% EtOH) for 15 min, washed 3× with $ddH_2O$, and mounted with 5% glycerol for visualization.

**Flow cytometry**. Cells were harvested in PBS and fixed in 70% EtOH for 30 min at 4 °C. Cells were washed 2× in PBS and incubated with 100 μg/mL of RNAse. Cells were incubated with 200 μl of propidium iodide (50 μg/mL) for 30 min, and cells were processed through a flow cytometer. The percentage of cells in the cell cycle was analyzed with the Cell Cycle platform using FlowJo software.

**ATP utilization assay**. Cells were grown in their respective condition for 48 h. To inhibit ATP production, 5 mM of 2-Deoxy-D-glucose (VWR) and 5 mM of NaAzide was added to cells for 20 min. ATP levels were measured using Promega's CellTiter-Glo® 2.0 Assay, following the manufacturer's protocol. ATP levels between cells treated with vehicle were compared to cells treated with drug at timepoints 0, 10, 40, and 60 after the initial 20-min incubation.

**Pulse SILAC (pSILAC)**. Cells were grown in light ($R_0K_0$) SILAC media (AthenaES) for 7 days and pulsed with heavy ($R_{10}K_8$) SILAC media (AthenaES) for 4 hr (MATRIX) or 16 h (TMT-pSILAC) following treatment.

**Ribosome-density fractionation**. Polyribosome fractionations were performed based on established protocol[20]. Briefly, cells were treated with 0.1 mg/ml of cycloheximide for the last 10 min of treatment, followed by ice-cold washes with

PBS−/− containing cycloheximide (0.1 mg/ml). Cells were then lysed in polysome lysis buffer (0.3 M NaCl, 15 mM MgCl$_2$·6H$_2$O, 15 mM Tris-HCl, pH 7.4, 1% Triton X-100, 0.1 mg/ml cycloheximide, and 100 units/ml RNase inhibitor). Following centrifugation (twice at 10,000×g for 5 min at 4 °C) to remove cellular debris, samples were loaded based on equal total RNA onto a 10–50% sucrose gradient and subjected to ultracentrifugation (187,813 × g for 1.5 h at 4 °C) using an SW 41 Ti rotor (Beckman Coulter). Samples were then fractionated into 1-ml fractions and collected using the BR-188 density-gradient fractionation system (Brandel). Total RNA was isolated from each fraction by phenol–chloroform extraction and ethanol precipitation following proteinase K treatment. Total protein was isolated by TCA precipitation (20% final TCA concentration), followed by three ice-cold acetone washes. Three independent experiments were pooled into a single sample for MATRIX MS analysis.

**MS analysis for MATRIX.** Samples were resuspended in 100 µL of 50 mM NH$_4$HCO$_3$ (pH 8.3), 8 M urea, and DTT was added to reduce cysteines at a final concentration of 10 mM. Cysteines were reduced at 60 °C for 1 h. The sample was cooled to room temperature, and iodoacetamide was added to a final volume of 20 mM. Samples were incubated at room temperature in the dark for 30 min. Samples were then acetone-precipitated overnight, and protein precipitates were centrifuged at 23,000g for 15 min. Precipitates were resuspended in 50 µL of NH$_4$HCO$_3$ (pH 8.3), and MS-grade Trypsin/LysC (Promega) was added to a final protease:protein ratio of 1:50, and samples were digested overnight at 37 °C. Samples were lyophilized and resuspended in 0.1% trifluoroacetic acid. Peptides were fractionated using the Pierce High pH Reverse Phase Peptide Fractionation Kit (Pierce), following the manufacturer's instructions. Each sample was fractionated into eight high-pH fractions.

Fractionated peptides were lyophilized, and lyophilized peptide mixtures were dissolved in 0.1% formic acid and loaded onto a 75-µm × 2-cm PepMap 100 Easy-Spray precolumn filled with 3 µm C18 beads (ThermoFisher Scientific) followed by an in-line 75-µm × 50-cm PepMap RSLC EASY-Spray column filled with 2 µm C18 beads (ThermoFisher Scientific) at a pressure of 700 BAR. Peptides were eluted over 120–240 min at a rate of 250 nl/min using a 0–35% acetonitrile gradient in 0.1% formic acid. For ribosome-density fractionated samples, free fractions were eluted over 120 min each, while 40/60/80 S and heavy-polysome fractions were eluted over 180 each. Samples were eluted over 240 min each. Peptides were introduced by nanoelectrospray into an LTQ-Orbitrap Elite hybrid mass spectrometer (ThermoFisher) outfitted with a nanospray source and EASY-nLC split-free nano-liquid chromatography (nano-LC) system (ThermoFisher Scientific). The instrument method consisted of one MS full scan (400–1500 m/z) in the Orbitrap mass analyzer, an automatic gain control target of 1e6 with a maximum ion injection of 200 ms, one microscan, and a resolution of 240,000. Ten data-dependent MS/MS scans were performed in the linear ion trap using the ten most intense ions at a normalized collision energy of 35. The MS and MS/MS scans were obtained in a parallel fashion. In MS/MS mode, automatic gain control targets were 1e5 with a maximum ion injection time of 50 ms. A minimum ion intensity of 5000 was required to trigger an MS/MS spectrum. Dynamic exclusion was applied using a maximum exclusion list of 500 with one repeat count with a repeat duration of 30 s and an exclusion duration of 15 s.

Raw MS files acquired from the mass spectrometer were processed using PEAKS software (Bioinformatics Solutions Inc.). Data were loaded into the software program, and data from each fraction were refined to merge scans within 2 min and 10.0 ppm. Spectra with PEAKS filter scores <0.5 were excluded. De novo sequencing and database searching was done using a precursor mass cutoff of 10.0 ppm and a fragment mass tolerance of 0.6 Da. Carbidomethylation of cysteine (+57.02 Da) residues was selected as a fixed modification, while variable modifications included 13C6-15N2 SILAC on K (8.01 Da), 13C6-15N4 SILAC on R (10.02), and Oxidation of M (15.99). Label-free quantification was performed in PEAKS using SILAC labels.

We filtered MS results in peptides found with greater than one unique peptide. For translation-factor enrichment, we filtered the results to canonical translation factors that were detected in all conditions (normoxia neutral, hypoxia neutral, and HA) and in every fraction (free, monosome, and polysome). The ratio of polysome to free was determined in every condition to estimate translational engagement. We considered a greater than twofold increase in acidosis compared to the other conditions as enriched in acidosis. We eliminated extreme outliers using the interquartile range method to remove potentially noisy data. To improve our confidence that candidates are indeed associated with actively translating ribosomes, we used the ratio of protein abundance in polysome fractions over 40S/60S/monosome fractions as a secondary readout to eliminate candidates that may be stalled at the initiation step of translation. We maintained consistency by using the standard cutoff of twofold increase compared to both basal and hypoxia-alone conditions (MATRIX_source file). Raw data available on ProteomeXchange: accession number PXD006799.

**MS analysis for TMT-pSILAC.** *TMT labeling and fractionation*: MS sample preparation and runs were performed by the SPARC Biocentre, The Hospital for Sick Children (Toronto, Canada). Samples were reduced, alkylated, digested, and TMT-labeled using the TMT10plex™ Isobaric Label Reagent Set (ThermoFisher Scientific, #90110) according to the manufacturer's directions. Labeled peptides from all

samples were combined and lyophilized. Peptides were then resuspended in 20 µl of ddH$_2$O and subjected to high-pH reversed-phase HPLC fractionation using a Waters XBridge C18 column. A 90-min gradient using buffer A (ddH$_2$O, adjusted to pH 10 with ammonium hydroxide) and buffer B (80% acetonitrile, adjusted to pH 10 with ammonium hydroxide) was run as follows: 0–3 min 1–12% B; 3–60 min 12–30% B; 60–65 min 30–60% B; 65–70 min 60–99% B; 70–75 min 99–1% B; 75–90 min 1% B. Ultraviolet (UV) absorbance was measured throughout the gradient at 214 and 280 nm using a Waters 2489 UV/visible detector. Fractions were collected from the beginning of the gradient in 1.2-min intervals for 60 fractions.

*MS analysis*: Fractionated samples were concatenated from 60 to 15 samples by mixing early, middle, and late fractions together. Samples were analyzed on an Orbitrap Fusion™ Lumos™ Tribrid™ Mass Spectrometer (ThermoFisher Scientific) outfitted with a nanospray and Evosep One LC system (Evosep). Lyophilized peptide mixtures were dissolved in 0.1% formic acid and loaded onto a C18 Evotip (Evosep). Samples were eluted and loaded onto a 15-C18 analytical column (100-µm ID, 3-µm beads) by Easy nLC1200 LC system (Thermo Scientific). A linear gradient of 0–42% buffer A (0.1% formic acid in water) to Buffer B (80% acetonitrile, 0.1% formic acid) was used with a 90-min run time. Data were acquired using the MultiNotch MS3 acquisition with synchronous precursor selection with a cycle time of 5 s. MS1 acquisition was performed with a scan range of 550–1800 m/z with the resolution set to 120,000, maximum injection time of 50 ms, and AGC target set to 4e5. Isolation for MS2 scans was performed in the quadrupole, with an isolation window of 0.6. MS2 scans were done in the linear ion trap with a maximum injection time of 50 ms and normalized collision energy of 35%. For MS3 scans, HCD was used, with a collision energy of 30%, and scans were measured in the orbitrap with a resolution of 50,000, a scan range of 100–500 m/z, an AGC Target of 3e4, and a maximum injection time of 50 ms. Dynamic exclusion was applied using a maximum exclusion list of 500 with one repeat count with an exclusion duration of 20 s.

*MS data analysis*: MS raw files were processed using Proteome Discoverer 2.2 (ThermoFisher Scientific). The MS data were searched against the Human Uniprot Database (downloaded April 10, 2017) consisting of only reviewed entries using the Sequest HT and MS Amanda 2.0 search engines. For both search algorithms, the parent and fragment mass tolerances were set to 10 ppm and 0.6 Da, respectively. Methionine oxidation was considered as a variable modification, as was N-terminal acetylation at the protein terminus. Static modifications of TMT at the peptide N terminus and carbamidomethylation of cysteines were also considered. When looking for all heavy-labeled proteins, fixed modifications of Heavy TMT (237,177 Da) on Lysine and Heavy 13C(6)15N(4) label on arginine were set. For all identifications, TMT and Heavy TMT were considered as dynamic modifications on lysine residues, as was heavy arginine. In each case, two missed cleavages were allowed. Search engine results were also processed through Percolator with q values set to 0.01 for strict and 0.05 for relaxed. TMT reporter ions were quantified using the Proteome Discoverer 2.2 reporter ion quantifier node with an integration tolerance of 20 ppm, on the MS order of MS3. Data were corrected for equal protein concentration used for TMT labeling and detection and cell number before analysis of fold change (correction factor HA: 0.650154799; HN: 0.80804954). The results were further filtered for proteins that were detected in every condition (NN, HN, and HA). The ratio of HA/NN and HA/HN was determined, and ratios greater than 1.5 (based on induction of glycolytic enzymes in HN vs. NN) were considered induced in acidosis (TMT-pSILAC_sourcefile). Raw data available on ProteomeXchange: accession number PXD015643.

**Immunohistochemistry.** Mouse CD1 whole-embryo sagittal paraffin sections (E12) were obtained from Zyagen. Paraffinized tumor core slides were obtained from Biomax. Slides were rehydrated using xylene/ethanol. Antigen retrieval was performed by boiling slides in citrate buffer (10 mM citric acid, 0.05% Tween, pH = 6.0) for 20 min, followed by cooling and washing in acetone for 1 min. Slides were blocked in 10% FBS for 3 h. Slides were incubated with primary antibodies (1:100) overnight at 37 °C. Slides were washed and incubated with secondary antibodies (1:200) for 2 h at 37 °C. Cells were washed and mounted with Invitrogen's Prolong Diamond Antifade Mountant. Antibodies used were HIF-1α (Novus, AF1935), CYR61 (GeneTex, N1C3), and PAI1 (Proteintech, 13801-1-AP).

**Mouse tumor xenograft assay.** All animal studies were performed under the approval of the University of Miami Institutional Animal Care and Use Committee (IACUC). Xenografts were generated through dorsal cell suspension injection (MCF7, 1.3e6 cells/mouse; 50% Matrigel in PBS) in NOD SCID gamma mice. MCF7 cells were subjected to HA conditions (1% O$_2$, pH 6.0) for 48 h prior to injection. Tumor volumes were measured by high-frequency ultrasound (Vevo 3100, VisualSonics). After three weeks, mice were euthanized, tumors collected, weighed, and processed for histological analysis.

**Immunoblot.** Sodium dodecyl sulfate polyacrylamide gel electrophoresis was performed on Bolt™ 4–12% Bis–Tris Plus premade gels (ThermoFisher Scientific) using the Mini Gel Tank system (ThermoFisher Scientific), and transferred to 0.2-µm Immuno-Blot® PVDF membranes (Bio-Rad) using the Bolt™ Mini Blot Module (ThermoFisher Scientific), all according to the manufacturer's protocols.

Chemiluminescent signals were detected using SuperSignal™ West Pico PLUS chemiluminescent substrate (ThermoFisher Scientific) on an Amersham Imager 600 (G9E Healthcare Life Sciences). The auto-capture function was used, whereby the machine performs a short pre-exposure to determine the optimal exposure time that yields the highest possible signal in the linear range of the camera below saturation. Densitometry was performed using ImageJ (NIH) to analyze gel function on 8-bit images that were in the linear range (i.e., had sharp and symmetrical peaks). Blots were stripped and reprobed when looking at total versus modified proteins unless otherwise noted. Antibodies (all 1:1000): GLUT1 (Novus Biologicals, # NB110-39113), PAI-1 (Proteintech, #13801-1-AP), Puromycin (3RH11) (Kerafast, #EQ0001), β-ACTIN (C4) (SantaCruz Biotechnology, #sc-47778), EIF5A (abcam, #ab32443), EIF3D (Proteintech, # 10219-1-AP), EIF3K (Proteintech, # 10640-1-AP), acetyl-EIF5A (Lys47) (Boster Bio, #P01727), CYR61 (Proteintech, #26689-1-AP), SDC4 (R&D Systems, #P31431), SIRT1 (Proteintech, #13161-1-AP), SIRT2 (Proteintech, #19655-1-AP), NFKB2 (Proteintech, #10409-2-AP), 4E-BP (Cell Signaling, #9452 S), Phospho-4E-BP (Ser65) (Cell Signaling, #9451 S), TSC2 (D93F12) (Cell Signaling, #4308S), gamma H2AX (Novus, NB100-384), H2AX (Abcam, Ab11175), CRM-1 (Novus, NB100-79802), XPO-4 (Abcam, ab133237), eIF5A2 (Thermo, #PA5-30770), Phospho-RAPTOR (Ser792) (Cell Signaling, #89146S), RAPTOR (Cell Signaling, #2280), Phospho-Tuberin/TSC2 (Ser1387) (Cell Signaling, #5584S), NDRG1 (Abcam, #ab37897), and Anti-hypusine (mAbHpu24) antibody was kindly provided by Genentech. The eIF5A1 antibodies and siRNAs used in this study were specific to eIF5A1 and not eIF5A2 as described by the manufacturer and validated in Supplementary Fig. 2f.

**RNA interference**. Target-specific pools of four independent siRNA) species (siGENOME SMARTpool, Dharmacon) were transfected at a final concentration of 50 nM using Effectene (Qiagen) for 48 h before treatment, according to the manufacturer's protocols.

**Nuclear/cytoplasmic extraction**. Cells were harvested in RNAse-free PBS and spun down. Cells were then resuspended in 0.1% NP40 to lyse the cytoplasm but keep nuclei intact. Cells were spun at 10,000$g$ for 10 s to pellet nuclei, and the cytoplasmic fraction was collected. Cells were washed 2× in 0.1% NP40. After the third wash, cells were resuspended in 0.1% NP40 and passed through a 25-g needle 50 times to lyse nuclei. RNA was isolated using TRIzol reagent following the manufacturer's instruction.

**Quantitative reverse transcription polymerase chain reaction**. First-strand cDNA synthesis was performed using the High-Capacity cDNA Reverse Transcription Kit (ThermoFisher Scientific), according to the manufacturer's protocols. Quantitative reverse transcription polymerase chain reaction was performed using the PowerUp™ SYBR® Green Master Mix (ThermoFisher Scientific) and a StepOnePlus™ Real-Time PCR System (ThermoFisher Scientific). Relative changes in expression were calculated using the comparative Ct ($\Delta\Delta$Ct) method.

**Primer sequences**. GAPDH (Forward 5′-CTGCACCACCAACTGCTT-3′; Reverse 5′-GTCTTCTGGGTGGCAGTG-3′), NDRG1 (Forward 5′-GCAGGCGCCTACATCCTAACT-3′); Reverse 5′-GCTTGGGTCCATCCTGA-GATCTT-3′), GLUT1 (Forward 5′-TGGCCGTGGGAGGAGCAGTG-3′; Reverse 5′-GCGGTGGACCCATGTCTGGTTG-3′), TSC2 (Forward 5′- TCACAGA-CAATGGGAGACACA-3′; Reverse 5′-CAAGTTCACCAGCACCAGAA-3′), Cyr61 (Forward 5′-AAGGAGCTGGGATTCGATGC-3′; Reverse 5′-CATTC-CAAAAACAGGGAGCCG-3′), Pai1 (Forward 5′-ACAACCCCACAGGAA-CAGTC-3′; Reverse 5′-GATGAAGGCGTCTTTCCCCA-3′), Sdc4 (Forward 5′-TGACTTTGAGCTGTCTGGCT-3′; Reverse 5′-GGTTATCTA-GAGGCACCAAGGG-3′), Lamb1 (Forward 5′- CCCCGGCTCTCCGTATGC-3′; 5′-TCTTCCCGTCTTCCTTTCCGGC-3′), Nfkb2 (Forward 5′- GGATC-CACGTCGACACCGTT; Reverse 5′-CCATCCAGACCTGGGTTGTAGC-3′), 18s (Forward 5′-CGCAGCTAGGAATAATGGAATAGG-3′; Reverse 5′-GCCTCAGTTCCGAAAACCA-3′), 5′ ETS (Forward 5′- TCTAGCGATCTGA-GAGGCGT; Reverse 5′-CAGCGCTACCATAACGGAGG-3′), and Junc (Forward 5′- GCCAACTCATGCTAACGCAG-3′; Reverse 5′-CTCTCCGTCGCAACTTGTCA-3′).

**Global protein synthesis measurement**. Global protein synthesis was measured by puromycin (ThermoFisher Scientific) incorporation (1 μg/ml final concentration for 20 min), followed by protein extraction using RIPA buffer (ThermoFisher Scientific) and immunoblot analysis with an anti-puromycin antibody (3RH11) (Kerafast, #EQ0001).

**Alkaline comet assay**. Comet assay protocol was adapted from Olive and Banath[110]. Cells were harvested after 48 h in condition and resuspended in 1% low-melting-point agarose (at 37 °C). Cells were lysed overnight in Alkaline Lysis (A1 lysis solution) solution supplemented with 1% triton. After lysis, cells were rinsed 3× in Alkaline rinse and electrophoresis solution (A2 alkaline rinse solution). Cells were then run at 4 °C for 30 min at 12 V (0.6 V/cm) in A2 rinse solution. Cells were neutralized in ddH$_2$O and stained with PI (2.5 μg/mL) in water for 20 min. Cells

were rinsed with water and mounted with an antifade solution. Comet tail analysis was done on ImageJ using the OpenComet plugin (opencomet plugin). H$_2$O$_2$ was used at 20 μM for 20 min after harvest as a positive control.

**TUNEL analysis**. Tunel assay was performed using the EZClick™ TUNEL kit (BioVision, #K191) following its manual. Pictures were taken on a microscope for red fluorescence generated by TUNEL-positive cells and green by total DNA, respectively. Nuclear intensity (positive staining) was measured compared to the cytoplasm on ImageJ.

**γH2AX staining**. Cells were fixed in cold 50% methanol/50% ethanol for 20 min at −20 °C. Cells were permeabilized with 0.5% Triton (PBS) and blocked in 5% FBS, 1% Triton for 60 min. Cells were stained with γH2AX (Novus, NB100-384) (1:50) for 2 h at 37 °C and secondary (1:500) for 1 h at 37 °C. Cells were washed, stained with Hoescht (1:1000), washed, and mounted. Foci were analyzed using FIJI's Find Maxima function.

**Fluorescent in situ hybridization (FISH)**. FISH was carried out with 5′ and 3′ digoxigenin (DIG)-labeled oligonucleotides. Post fixation of 30′, cells were quenched with 0.1 M Tris-HCl, pH 7.0 for 10′ before ± Proteinase K (PK) treatment (NEB, 800 U/ml stock, 100,000× dilution) at 37 °C for 30′. Cells were equilibrated in 2× SSC before O/N hybridization at 37 °C. Probes (10 pmol) were denatured at 85 °C for 10′. Hybridization buffer was 15% formamide, 10% dextran sulfate, 2 mM vanadyl ribonucleoside, and 2× SSC. Probes were detected with an anti-DIG-Fluorescein antibody (Sigma, 11207750910) at 20 μg/ml in 4× SSC. Slides were mounted in 90% glycerol.

Tsc2 mRNA (4666-4715): 5′CAGAGAAAGTGCCAGGCATCAACCCC AGTTTCGTGTTCCTGCAGCTCTAC 3′

Antisense Probe sequence:
5′: GTAGAGCTGCAGGAACACGAAACTGGGGTTGATGCCT GGCACTTTCTCTG 3′

**RNA immunoprecipiation**. RNA immunoprecipitation was done using the RiboCluster Profiler RIP-Assay Kit (MBL) following its manual using ChiP-Grade Protein A/G Magnetic Beads (Thermo, Prod #26162). eIF5A-RNA was pulled down using 5 μg of anti-eIF5A (SantaCruz, sc-390202), and an equal amount of mouse IgG was used as a control.

**Statistical analysis**. All experiments were performed at least three independent times unless otherwise stated. Quantitation of microscopy-based data was performed on at least five representative images. Appropriate statistical analyses were performed, e.g., Student's $t$ test, Mann–Whitney, and chi-square test. Statistical significance was defined as $p < 0.05$.

**Reporting summary**. Further information on research design is available in the Nature Research Reporting Summary linked to this article.

## Data availability

The data that support this study are available from the corresponding author upon reasonable request. Mass spectrometry datasets are available via the ProteomeXchange accessions: PXD006799 and PXD015643. The mass spectrometry proteomics data have been deposited to the ProteomeXchange Consortium via the PRIDE[111] partner repository with the dataset identifier PXD015643 and PXD006799. MS data were searched against the Human Uniprot Database (https://www.uniprot.org) consisting of only reviewed entries using the Sequest HT and MS Amanda 2.0 search engines. Source data are provided with this paper.

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

## Acknowledgements

N.C.B. is the recipient of an NIH F30 Fellowship grant (CA243268-01). J.J.D.H. is the recipient of a Canadian Institutes of Health Research (CIHR) Postdoctoral Fellowship. J.H.S. is funded by grants from the NIH/NCI (5R01CA190696 and 5UM1CA154967), the Florida Academic Cancer Center Alliance, and the Sylvester Comprehensive Cancer Center. S.L. is funded by grants from the NIH (NIGMS, 1R01GM115342, and NCI, 1R01CA200676) and the Sylvester Comprehensive Cancer Center (SCCC). We thank Siôn Ll. Williams and the SCCC Oncogenomics Core Facility for RNA-sequencing services.

## Author contributions

N.C.B., J.J.D.H., and S.L. conceptualized the study and designed the experiments. N.C.B., J.J.D.H., P.R.T., M.H.B., M.W., L.M.L., and J.R.K. performed the experiments. J.H.S. and S.L. provided the reagents and resources for experiments. N.C.B., J.J.D.H., and S.L. analyzed the data and wrote the paper.

## Competing interests

The authors declare no competing interests.
