## [Peer Review File · Nature Communications]

Reviewers' comments:

Reviewer #1 (Remarks to the Author):

This study by Balukoff and colleagues identify and characterise an adaptive translational pathways that enable cells to adapt to anaerobic metabolism-induced extracellular acidosis, a frequently encountered stress encountered in health and disease. During acidosis, like many cellular stresses, cells will globally repress energy expensive processes, such as bulk protein synthesis, but must still actively synthesise protective factors that can respond to the insult. In this study the authors Identified eIF5A as a key factor that controls this specific translational program and demonstrated that this pathway is important for maintaining genomic stability during stress conditions.

Specifically, a mass spectrometry screen of translation factors was performed to identify factors enriched in the actively translating fraction of cells encountering hypoxia acidosis. eIF5A, was identified to be specifically enriched under these conditions and subsequent depletion of eIF5A in cell culture lines reversed the acidosis-dependent translational suppression. Tsc2 mRNA interacts with eIF5A and is a key translational target. Furthermore, depletion of Tsc2 was sufficient to phenocopy eIF5A depletion. Depletion of eIF5A and Tsc2 reversed the acidosis-dependent acidosis phenotype, however this caused increased DNA damage, suggesting that the eIF5a/ Tsc2 pathway preserves genome integrity in response to acidosis stress.

This is a well performed and interesting study, with robust and novel data and a clear message. However, the study could be improved with the following suggestions.

Specific Comments

1. The western blots in Figure 1 are over exposed and unsuitable for quantification. Also, not clear if the intensities are means of the repeats or quantitation of the representative blots.
2. In Figure 2A-C. Does depletion of eIF5A rescue the ATP/ transcription/ translation phenotype to the levels seen in neutral cells. Also, does eIF5A depletion in neutral cells have any effect on these processes.
3. RNAi experiments were done using smart pools of 4 RNAi's against each protein. Single RNAi's should be used to confirm the key findings to ensure there are no off-target effects.
4. Please provide separate blots for Figure 4A, or make it clear in the figure legend the eIF5A, actin and puromycin panels are re used from Figure 2C.
5. In Figure 5 measurement of gamma H2AX foci is a much more quantitative way of measuring DNA damage than immunoblot analysis and would provide quantitative, rather than a qualitative readout.
6. The link between Sirt 1 and eIF5A is not clear in the study. Specifically, the data in Figure 7 demonstrating the involvement of Sirt1 in EIF5A deacetylation is unclear and messy and relies on a single chemical inhibitor. Better immunoblots are needed along with a Sirt1 RNAi to complement chemical inhibition. Also does Sirt1 directly interact with eIF5A, either in vitro or by co-IP.

Reviewer #2 (Remarks to the Author):

The paper by Balukoff et al describes the role of eIF5A as a specific regulator of the translome during acidic hypoxia. The protein was initially 'discovered' using a proteomics screen involving pulsed SILAC and density fractionation. Subsequently, the authors hypothesise that Sirt1 acts upstream of eIF5A, through a wide range of experiments. They also concluded that Sirt1 is causing the deacetylation of eIF5A and subsequent re-localization to the cytoplasm only during acidic hypoxia. Interesting they put forward a model where eIF5A mode of action is not through boosting

the efficiency of translation elongation of the target mRNA, i.e. tsc2, but through enhancing its nuclear export. Although the authors have clearly shown the importance of eIF5A in acidic hypoxia and generated quite a body of work from which they draw their conclusions unfortunately, the work's proposed relationships for Sirt1, Tsc2 and mTORC1 are still based on correlations rather than direct evidence. Furthermore, the underpinning proteomics data needs to be clarified and made statistically robust.

Comments

1- The experimental design of the proteomics work is quite powerful and provides insight into hypoxic conditions. However, the experiments appear to be performed once and the data analysis appears rudimentary. The presentation appears to involve simply ranking fold change of proteins, choosing an arbitrary fold change cut off and then making subjective (not necessarily wrong) choices for protein targets to take forward. The authors need to provide far more detail (including supp tables) and clearer statistical rationale for their choices.

2- The author applied two distinct MS search strategies for the TMT-pSILAC and Matrix experiments, Proteome Discoverer (search against human database) and Peaks (with denovo search). Can the authors explain their choices? The two disparate approaches will affect consistency of analysis.

3- The authors very clearly show that Sirt1 plays a role in eIF5A deacetylation; however, the data is correlative, and the relationship may not be direct. The work would benefit from data showing a clearer relationship e.g. direct evidence of Sirt1- eIF5A having an enzyme-substrate relationship.

4- The model where eIF5A re-localization causing the cytoplasmic export of Tsc2 mRNA is built on correlation and negative data. For this model to be true, eIF5A needs to have very strong affinity and clear binding motif on Tsc2 mRNA, to avoid exporting unrelated mRNAs under stress. The authors showed that this motif is not in the coding sequence but that is limited evidence regarding specificity. The authors could ascertain clearer evidence using high through-put approaches such as HITS-CLIP to demonstrate binding specificity and distribution of binding sites. Alternatively, they could locate and clone the eIF5A-responsive element from Tsc2 mRNA in reporter gene and reproduce the cytoplasmic export effect.

There is much merit in the manuscript, and it would be acceptable for publication if the above issues were addressed.

Reviewer #3 (Remarks to the Author):

In this article, Balukoff et al provide evidence suggesting a hitherto unappreciated mechanism of adaptation to hypoxic acidosis. The proposed model encompasses Sirt1-dependent deacetylation and relocalization of eIF5A from the nucleus into the cytoplasm. This is accompanied by upregulation of TSC2 (via increased nuclear export and translation of corresponding mRNA), inhibition of mTORC1, and reduction in global translation and cell proliferation. Finally, it is proposed that this mechanism sustains genome integrity in acidosis. Although I found this study to be of a high potential interest, several outstanding issues with scientific premises, methodology and interpretation of the results were observed. At least in this reviewer's opinion addressing these issues should be considered in order to support authors' conclusions. My specific comments and concerns are outlined below.

Major concerns:

- "These translation factors operate as translome remodelers (which we define as polysome-associated proteins that regulate translation intensity via post-transcriptional mechanisms) that

control protein outputs to activate biochemical pathways.” This statement is arguably inaccurate which brings the validity of the author’s method (MATRIX) into question. The vast majority of translation initiation factors do not associate with polysomes (although some were reported to do so, but in a rather non-reproducible manner). What they remodel are translation initiation complexes, whereby translome remodeling is commonly defined as changes in the association of subset of mRNAs, and not translation factors, with the polysomes between two or more conditions (e.g. PMID: 29959195).

-Related to above, I am not familiar with the evidence that accumulation of translation initiation factors in polysomal fractions is indicative of productive translation, and thus I thought that the following statement should be corroborated with some references: “Specifically, translation factors, and mRNAs undergoing productive translation, accumulate in polysome fractions, while cellular assets disengaged from active protein synthesis are relatively enriched in the free fractions”.

-Finally, notwithstanding that eIF5A likely has a major role in the early steps of elongation, it is not clear why “polysomal enrichment” of translation initiation factors is used as a criteria for remodeling of translation initiation machinery. The authors should use shallower gradients/crosslinking +/- nonhydrolyzable GTP to capture translation initiation complexes, which is a standard in the field.

-“These findings highlight the generality of the impact eIF5A exerts during anaerobic acidosis, in agreement with its reported role as a tumor suppressor”. The authors should explain the potential reasons for the discrepancies between their findings that eIF5A depletion does not affect cell proliferation under base-line conditions and stimulates tumor growth and a number of studies showing the opposite (e.g. PMID: 29321164, 15205331, 30761741, 24220243) while supporting the tumor promoting role for eIF5A isoforms (26483550, 11325856, 14622290, 15205331)

- Related to the above, the most of experiments (including those in vivo) were carried out using siRNA. What is the evidence that eIF5A levels were suppressed across the course of the study? More stable depletion strategies as well as the appropriate rescue experiments should be considered.

-Stronger evidence that nuclear export of eIF5A should be provided. Using eIF5A mutants which are restrained to the nucleus should be preferred over just depleting eIF5A levels, to support the authors’ claims.

- TSC2-dependent suppression of mTOR signaling under acidosis was reported previously (21738705) and this study should be referenced by the authors. It is also surprising that authors did not exclude alternative mechanisms of TSC-dependent mTOR inactivation in acidosis (e.g. AMPK), which I thought was warranted to support their conclusions.

-Term “metabolic depression” seems inappropriate. The authors should consider using “metabolic reprogramming” or a similar term.

-Figure 7 – Sirt1 depletion (KO or KD) condition would be useful to support the Sirt1 inhibitor (Ex-572) data

- In Fig 2.i,j) is n=6 sufficient to draw these conclusions? What is the power of this study? Related to the above, these studies were carried out with siRNA. What is the evidence that eIF5A was still depleted during the course of the experiment?

-Statistical analyses are not clearly described (there is no description which tests were used in the figures). Some additional issues with statistical analyses were also observed. For instance, it is more appropriate to use SD than SEM for e.g. densitometry measurements. It is also not clear why is densitometry used in some cases but not the others. Moreover, if controls were done in triplicate the authors should propagate SD values and present them on graphs.

-Only evidence of metabolic reprogramming comes from the levels of enzymes involved in the metabolic pathways and not direct measurements of metabolites in these pathways. Therefore, the authors should either perform adequate measurements of metabolites or tone down their claims.

-It is not clear which of the eIF5A isoforms is implicated in the observed phenomena (eIF5A1 or 2)?

Minor comments:

-In supplementary figure 1a the colors of lines are very similar and hard to distinguish. I appreciate that the colors are chosen to represent “hypoxia neutral pH” and “hypoxia acidosis” states throughout the manuscript, but in this particular case the authors are encouraged to e.g.

use a broken line for one of the conditions.

-It should be clarified what was the precise criteria for the distinction between what is referred to as oligosomes vs. polysomes. There is already enough confusion in the field, so the authors are encouraged to use simpler terms such as light and heavy polysomes.

-In figure 5b the number of replicates is not indicated.

-In supp. figure 6A (WI-38 normoxia neutral pH condition) DAPI does not appear to correspond to eIF5A staining. Authors should consider showing panel overlays.

-The authors state "This contrasts with the role of Sirt2 as a major deacetylase of eIF5A under basal conditions" indicating the role of Sirt2 as a deacetylase of eIF5A in basal conditions.

However, in Supplementary Fig 6. c) with the use of an Sirt2 inhibitor (AGK2) there is no marked rise in eIF5A acetylation.

-Work from Tauc's group shows that in kidney suppression of eIF5A activity (by targeting hypusination) prevents cell death under anoxia by silencing mitochondrial functions. This work should be commented on.

I hope that the authors will find this criticism constructive and that my assessment of their work will hold sufficient pathos.

Sincerely

I/Topisirovic

Reviewer #1

We are delighted that Reviewer #1 believes that our study is interesting, with robust, novel data and a clear message. We thank Reviewer #1 for the valuable and constructive critiques, which we have answered below.

Comment #1. The western blots in Figure 1 are over exposed and unsuitable for quantification. Also, not clear if the intensities are means of the repeats or quantitation of the representative blots.

Response. We now provide lower exposures of western blots for Fig. 1. The intensities are means of repeats and the measurement data have been moved to Supplementary Figures.

New data.

New Fig. 1f. Lower exposure of previous Fig. 1f.

Comment #2. In Figure 2A-C. Does depletion of eIF5A rescue the ATP/ transcription/ translation phenotype to the levels seen in neutral cells. Also, does eIF5A depletion in neutral cells have any effect on these processes.

Response. Depletion of eIF5A increases translation intensity by about 3-fold in hypoxic/acidotic conditions whereas it has modest effects in neutral cells. While silencing of eIF5A does not rescue the translation and transcription phenotype to level seen in neutral cells, it should be noted that the increased activity is sufficient to sustain cellular proliferation and viability under hypoxic/acidotic conditions. We have added these new datasets to Supplementary Fig. 3a-b and modified the text accordingly.

New data.

New Supplementary Fig. 3a. Transcription Intensity in eIF5A siRNA cells New Supplementary Fig. 3b Translation Intensity in eIF5A siRNA cells

New Text

Page 7. Results revealed that eIF5A knockdown led to a significant resumption of ATP utilization (**Fig. 2a**), transcription (**Fig. 2b**), and protein synthesis (**Fig. 2c, Supplementary Fig 3c**) in anaerobic acidotic cells but had only relatively modest effects in cells maintained in neutral conditions (**Supplementary Fig. 3a-b**).

Page 7. Silencing eIF5A in hypoxic/acidotic conditions did not completely restore transcription/translation activity to levels seen in neutral cells but was sufficient to sustain cellular proliferation and viability (**Supplementary Fig. 3a-b**).

Comment #3. RNAi experiments were done using smart pools of 4 RNAi's against each protein. Single RNAi's should be used to confirm the key findings to ensure there are no off-target effects.

Response. This is an excellent point raised by the Reviewer. We have reproduced experiments with single RNAi's targeting and obtained similar results for eIF5A/Tsc2.

New data.

New Supplementary Fig. 5c. Effect of single siRNA against eIF5A on Tsc2.

New Text.

Page 8. During hypoxia acidosis, eIF5A knockdown by pooled or single-site siRNA attenuates Tsc2 protein induction and de-represses mTORC1 activity, resulting in increased 4E-BP phosphorylation and enhanced global translational intensity (**Fig. 4a-b, Supplementary Fig. 5a-c, Fig. 2c**)

Comment #4. Please provide separate blots for Figure 4A, or make it clear in the figure legend the eIF5A, actin and puromycin panels are re used from Figure 2C.

Response. We sincerely apologize for this mistake. We have reproduced both Figure 2C and Figure 4A with new experiments and added these new blots to the paper.

New Data.

New Fig. 2C. Effect of silencing eIF5A on puromycin incorporation.

New Fig. 4B. Effect of silencing eIF5A on puromycin incorporation

Comment #5. In Figure 5 measurement of gamma H2AX foci is a much more quantitative way of measuring DNA damage than immunoblot analysis and would provide quantitative, rather than a qualitative readout.

Response. We agree with the Reviewer and now include immunofluorescence analysis to detect foci of gamma H2AX in normoxia neutral, hypoxia neutral, and hypoxia acidosis (siRNA control and eIF5A siRNA) as well as hypoxia/acidosis (siRNA control, eIF5A siRNA, Tsc2 siRNA, and eIF5A siRNA with Torin 2). These experiments are consistent with the immunoblots.

New Data

New Fig. 5c. Immunofluorescence analysis of H2AX foci.

New Fig. 5e. Immunofluorescence analysis of H2AX foci.

New Text.

Page 10. Indeed, eIF5A-competent cells effectively maintained genomic integrity during acidosis, as determined by DNA damage measurements using alkaline comet (Fig. 5a) and TUNEL (Fig. 5b) assays, as well as γ H2AX foci formation and levels⁷⁰ (Fig. 5c, Supplementary Fig. 6a).

Comment #6. The link between Sirt 1 and eIF5A is not clear in the study. Specifically, the data in Figure 7 demonstrating the involvement of Sirt1 in EIF5A deacetylation is unclear and messy and relies on a single chemical inhibitor. Better immunoblots are needed along with a Sirt1 RNAi to complement chemical inhibition. Also does Sirt1 directly interact with eIF5A, either in vitro or by co-IP.

Response. As per the request of the Reviewer, we have performed siRNA-mediated silencing of Sirt1 and obtained similar results compared to the chemical inhibitor Ex-527. In addition, we have added a reference (PMID22771473: data shown in their Supplementary Fig.1a) that shows that Sirt1 can deacetylate eIF5A in vitro. We hope that the new siRNA datasets and previous work from another group will satisfy the concerns of the Reviewer.

New Data

New Fig. 7d. Effect of silencing Sirt1 on eIF5A acetylation.

New Text

Page 12. Sirt1 and Sirt2 are two major eIF5A deacetylases in cells and in enzyme/substrate in vitro assays⁴¹ that are regulated by NAD^+ , a principal product of fermentation^{75,764}. We found that Sirt1 and Sirt2 specifically deacetylates eIF5A during anaerobic acidosis and in neutral conditions, respectively (Fig. 7c-d, Supplementary Fig. 8e).

Reviewer #2

We are pleased that the Reviewer has highlighted the merit of our manuscript, and that it would be acceptable for publication if the issues raised were addressed. We thank the Reviewer for proposing experiments to solidify the link between Sirt1, Tsc2 and mTORC1, which we have answered below.

Comment #1. The experimental design of the proteomics work is quite powerful and provides insight into hypoxic conditions. However, the experiments appear to be performed once and the data analysis appears rudimentary. The presentation appears to involve simply ranking fold change of proteins, choosing an arbitrary fold change cut off and then making subjective (not necessarily wrong) choices for protein targets to take forward. The authors need to provide far more detail (including supp tables) and clearer statistical rationale for their choices.

Response. As per the request of the Reviewer, we include Source Tables with raw MS datasets in the revised version of the manuscript. For the MATRIX, we selected eIF5A since it was the only translation factor that displayed an increase in translating ribosomes in hypoxic/acidotic cells compared to both normoxic/neutral and hypoxic/neutral. As we successfully validated the MATRIX, we opted to pursue our studies focusing on eIF5A, especially since this translation factor controls the cellular dormancy phenotype during anaerobic acidosis.

For the translome MS, we agree with the Reviewer that we may have overstated our conclusions and have considerably re-worked the text and changed Figure 3 as indicated below to indicate that we used the TMT-pSILAC datasets as a screen for detection of targets in the eIF5A mediated dormancy pathway. Briefly, we initially selected candidates as follows: 1- previously published results by other groups showing acidosis-induced expression (e.g. Pai1 and Cyr61 (PMID: 16736314, 21447598) to validate our MS datasets, 2- candidates that displayed various fold-enrichment above a threshold of 1.5, a level that is used in other papers (PMID:30394099, 15314609), and 3- the availability of high-quality commercial reagents. Tsc2/Sirt1 were selected via a screen of the TMT-pSILAC to identify potential candidates that could possibly explain the eIF5A/anaerobic metabolism-induced metabolic depression. After validation of the MS targets, we decided to focus on the Sirt1/eIF5A/Tsc2 axis at the center of this paper. These strategies are now better explained in the text. We hope that this will satisfy the Reviewer.

Deleted Text and Original Figures.

Page 8. Analyses revealed a global reorganization of protein output in response to hypoxia acidosis, compared to both normoxia or hypoxia neutral pH conditions (**Fig. 3a, 3b**).

Page 8. A systems-level assessment of mRNA translation efficiency, defined as the ratio of polysomal abundance to combined monosomal and free abundance (**Supplementary Fig. 2a**) and steady-state level (aggregate abundance of all fractions) changes (**Supplementary Fig. 4b**) showed that the induction of these 244 proteins are mediated predominantly through translation efficiency regulation (**Fig. 3c, right panel**). Concordance analysis further revealed translation efficiency as a superior predictor of protein output compared to steady-state mRNA level fluctuations in anaerobic acidotic cells (**Supplementary Fig. 4c**).

Page 9. Immunoblot validation of representative targets identified by TMT-pSILAC verified potentially adaptive proteins/markers of the hypoxia-induced acidotic state (**Fig. 3h**).

Page 9. Consistent with acidosis-induced metabolic depression (**Fig. 2, Supplementary Fig. 3**), pathway analysis of the anaerobic acidosis translome (TMT-pSILAC) revealed a down-regulation of major metabolic processes during acidosis (**Fig. 3d**) and up-regulation of effectors that mediate exit from mitosis and RNA synthesis termination (top two enriched processes) (**Fig. 3e**). Confirming published observations^{14,15,47-49}, our analysis identified an acidosis-dependent reduction in glycolytic intensity (**Fig. 3f**), in contrast to other metabolic pathways e.g. glutaminolysis and oxidative phosphorylation (**Fig. 3g**).

New Text.

Page 8. Well-characterized candidates uncovered by TMT-pSILAC and that displayed various fold-enrichment above a threshold of 1.5^{50,51} were validated by immunoblot blot analysis (**Fig. 3d, Supplementary Fig 3c**).

Comment #2. The author applied two distinct MS search strategies for the TMT-pSILAC and Matrix experiments, Proteome Discoverer (search against human database) and Peaks (with denovo search). Can the authors explain their choices? The two disparate approaches will affect consistency of analysis.

Response. We believe that PEAKS performs in a superior manner for SILAC data. Proteome Discoverer was used for the TMT-SILAC data because at the time, PEAKS did not support TMT data all that well. We have internally done a comparison between PEAKS and Proteome Discoverer and have found the identifications and quantification for the TMT-SILAC data to be constant. For consistency across the study, we decided to keep the search engines as described in the original version of the paper.

Comment #3. The authors very clearly show that Sirt1 plays a role in eIF5A deacetylation; however, the data is correlative, and the relationship may not be direct. The work would benefit from data showing a clearer relationship e.g. direct evidence of Sirt1- eIF5A having an enzyme-substrate relationship.

Response. This is a good comment by the Reviewer. Direct evidence of Sirt1- eIF5A having an enzyme-substrate relationship was reported in PMID:22771473 (in their supplemental fig S1). We have included this citation in the new version of the paper. Nonetheless, to answer the question of the Reviewer, we silenced Sirt1 with siRNA to demonstrate an effect on eIF5A acetylation status during acidosis, in agreement with data obtained with the chemical inhibitor Ex-527.

New Data.

New Fig. 7d. Effect of silencing Sirt1 on eIF5A acetylation.

New Text

Page 12. Sirt1 and Sirt2 are two major eIF5A deacetylases in cells and in enzyme/substrate in vitro assays⁴¹ that are regulated by NAD⁺, a principal product of fermentation^{75,76}. We found that Sirt1 and Sirt2 specifically deacetylates eIF5A during anaerobic acidosis and in neutral conditions, respectively (**Fig. 7c-d, Supplementary Fig. 8e**).

Comment #4. The model where eIF5A re-localization causing the cytoplasmic export of Tsc2 mRNA is built on correlation and negative data. For this model to be true, eIF5A needs to have very strong affinity and clear binding motif on Tsc2 mRNA, to avoid exporting unrelated mRNAs under stress. The authors showed that this motif is not in the coding sequence but that is limited evidence regarding specificity. The authors could ascertain clearer evidence using high throughput approaches such as HITS-CLIP to demonstrate binding specificity and distribution of binding sites. Alternatively, they could locate and clone the eIF5A-responsive element from Tsc2 mRNA in reporter gene and reproduce the cytoplasmic export effect.

Response. To address the question regarding specificity of eIF5A/Tsc2 mRNA interaction, we performed new RNA immunoprecipitation (RIP), nuclear/cytoplasmic distribution and translation efficiency experiments followed by qPCR with primer sets targeting various mRNAs. As shown in the new **Fig. 5d**, eIF5A specifically binds to Tsc2 mRNA as well as to other eIF5A-regulated targets (c-Jun and Scd4) (Fig. 5j) but not to various eIF5A- or acidosis-independent mRNAs. In addition, eIF5A specifically regulates acidosis-dependent nuclear export and translation efficiency of Tsc2, c-Jun and Scd4 mRNAs but not eIF5A-independent mRNAs. These new data provide evidence supporting our model of specific interactions between eIF5A and its regulated mRNAs

(Fig. 6a) to control nuclear export (Fig. 6b) and translation efficiency (Fig. 6c) while it does not bind or regulate eIF5A-independent mRNAs.

As for the 5' and 3' UTR experiments, there are at minimum >12 different UTRs in Tsc2 mRNA that have been described in the literature limiting our ability to address this question in a timely manner. It should be noted that other groups have not reported a direct interaction between eIF5A and RNA and that eIF5A probably participates in multi RNP complexes, which may be difficult to recapitulate in overexpression experiments (or perform HITS-CLIP). We have added the new data and changed the text accordingly.

New Data.

New Fig. 6a. RNA immunoprecipitation of eIF5A

New Fig. 6b. Effect of silencing eIF5A on the nuclear/cytoplasmic distribution of mRNAs during anaerobic acidosis.

New Fig. 6c. Effect of silencing eIF5A on translation efficiency of mRNAs during anaerobic acidosis.

New Text

Page 11. Next, we examined the mechanisms by which eIF5A controls Tsc2 protein induction during anaerobic acidosis. RNA immunoprecipitation experiments revealed that eIF5A specifically associated with the mRNAs of Tsc2 and two other eIF5A-regulated proteins c-Jun and Scd4 but not the eIF5A-independent Cyr61, or control transcripts (**Fig. 6a, Supplementary Fig. 7a-b**). Consistent with these results, nuclear export of Tsc2, c-Jun and Scd4 (**Fig. 6b-c, Supplementary Fig. 7c**) mRNAs during anaerobic acidosis was dependent on eIF5A, resulting in increased engagement by cytoplasmic polysomes (**Fig. 6d**) without significantly affecting steady-state mRNA expression (**Supplementary Fig 7d**). In stark contrast, eIF5A depletion had no effect on nuclear export and polysome engagement of eIF5A-independent mRNAs (**Fig. 6b, Fig 6d, Supplementary Fig. 7d**). These results provided evidence that eIF5A specifically interacts with its regulated mRNAs, but not other transcripts, to control their nuclear export and translation efficiency. In agreement with this, treatment with leptomycin B, which inhibits eIF5A nuclear export^{71,72} (**Fig. 6e**) prevented anaerobic acidosis-induced cytoplasmic export of Tsc2 mRNA (**Fig. 6f**) reducing Tsc2 protein levels (**Fig. 6g**).

Reviewer #3.

We are happy that Reviewer #3 believes that our study is of a high potential interest and thank him/her for the valuable and constructive comments, which we have answered below.

Comment #1. “These translation factors operate as translome remodelers (which we define as polysome-associated proteins that regulate translation intensity via post-transcriptional mechanisms) that control protein outputs to activate biochemical pathways.” This statement is arguably inaccurate which brings the validity of the author’s method (MATRIX) into question. The vast majority of translation initiation factors do not associate with polysomes (although some were reported to do so, but in a rather non-reproducible manner). What they remodel are translation initiation complexes, whereby translome remodeling is commonly defined as changes in the association of subset of mRNAs, and not translation factors, with the polysomes between two or more conditions (e.g. PMID: 29959195).

Related to above, I am not familiar with the evidence that accumulation of translation initiation factors in polysomal fractions is indicative of productive translation, and thus I thought that the following statement should be corroborated with some references: “Specifically, translation factors, and mRNAs undergoing productive translation, accumulate in polysome fractions, while cellular assets disengaged from active protein synthesis are relatively enriched in the free fractions”.

Finally, notwithstanding that eIF5A likely has a major role in the early steps of elongation, it is not clear why “polysomal enrichment” of translation initiation factors is used as a criterion for remodeling of translation initiation machinery. The authors should use shallower gradients/crosslinking +/- nonhydrolyzable GTP to capture translation initiation complexes, which is a standard in the field.

Response. We agree with the Reviewer and have extensively modified the text (please see below) to address these comments, better explain our strategy and temper down our initial claim regarding MATRIX methodology. Indeed, the polysomes/free or polysomes/monosomes would be predicted to have a bias toward selecting for translation elongation factors or other proteins associated with translating mRNAs. Nonetheless, there are papers (e.g. PMID: 17237356, 23716667, 28460002, 30095066, 22678294, 29298419) that have shown the presence of translation initiation factors in polysomes. During anaerobic acidosis, eIF5A is likely associated with translating mRNAs explaining why it was detected by MATRIX as enriched in the polysomes fractions. We hope that these modifications will satisfy the Reviewer.

Modified Text (~~deleted text~~, new text)

Page 4. This hypothesis was tested using several systems-level technologies, including our recently developed MATRIX (mass spectrometry analysis of ~~active~~ translation factors using ribosome density fractionation and isotopic labeling experiments) platform, which generates snapshots of translation factor ~~activity~~ distribution in free, monosomes, light and heavy polysomes fractions under different cellular conditions.

Page 5. ...our recently developed MATRIX platform (mass spectrometry analysis of ~~active~~ translation factors using ribosome density fractionation and isotopic labeling experiments), which ~~discriminates~~ identifies factors based on ~~translational activity~~ their distribution in sucrose gradients (e.g. free, monosomes, light and heavy polysomes fractions)

Page 6. Grey columns indicate translation factors whose ~~activities~~ distribution are not affected by anaerobic acidosis, while red (Fig. 1b) and yellow (Fig. 1c) columns represent translation factors that are relatively enriched in ~~activity~~ polysomes under basal and hypoxia neutral pH conditions, respectively.

Page.6 We focused on eIF5A as it represented the translation factor that displayed an increase in ~~translational~~ heavy polysomes engagement under hypoxia acidosis conditions specifically, compared to both basal (Fig. 1b) and hypoxia neutral pH conditions (Fig. 1c). For instance, we did not study eIF4B as it also accumulates in heavy polysomes under hypoxia neutral pH conditions (Fig. 1c). Analysis using the ratio of heavy polysome/monosome protein abundance as a secondary readout confirmed the enrichment of eIF5A ~~activity~~ in heavy polysomes specifically under anaerobic acidosis (Supplementary Fig. 2b, c).

Page 3. These translation factors operate as translome remodelers (~~which we define as polysome-associated proteins that regulate translation intensity via post-transcriptional mechanisms~~) that control protein outputs at the translational level to activate biochemical pathways.

New Text.

Page. 5. MATRIX has an inherent bias toward translation elongation factors associated with heavy polysomes although translation initiation factors are also detectable in the heavy polysomic fractions under various experimental settings^{20,24,36-38}.

Deleted Text

Page 5. Specifically, translation factors, and mRNAs undergoing productive translation, accumulate in polysome fractions, while cellular assets disengaged from active protein synthesis are relatively enriched in the free fractions (**Fig. 1a**).

Comment #2. “These findings highlight the generality of the impact eIF5A exerts during anaerobic acidosis, in agreement with its reported role as a tumor suppressor”. The authors should explain the potential reasons for the discrepancies between their findings that eIF5A depletion does not affect cell proliferation under base-line conditions and stimulates tumor growth and a number of studies showing the opposite (e.g. PMID: 29321164, 15205331, 30761741, 24220243) while supporting the tumor promoting role for eIF5A isoforms (26483550, 11325856, 14622290, 15205331).

Response. We have modified the text to consider that various laboratories have obtain seemingly opposite results following silencing of eIF5A. In our hands, we did not notice any significant effects on growth and viability with MCF7, U87MG and WI-37 cells under normoxia/neutral and hypoxia/neutral conditions. We have also clarified in the paper that we are specifically looking at the effect of eIF5A1 knockdown (see response to Comment 11). It should be noted that other have

shown that eIF5A was growth limiting only in cells that harbored Kras mutations and silencing of eIF5A had no effect on growth of MEFs, HUVEC and several cancer cell lines (PMID: 29321164). In addition, others have only reported a modest effect after prolonged incubation time (PMID: 24220243, 22927971) while eIF5A knockdown had no effect on proliferation of colon cancer cells (PMID: 17187778). Finally, these studies were assessing the role of eIF5A in normoxia-neutral pH (basal) conditions.

We have modified the text as follows.

Modified Text (~~deleted text~~, new text)

Page 7. line These findings highlight the generality of the impact eIF5A exerts during anaerobic acidosis, ~~in agreement with its reported role as a tumor suppressor~~⁴²¹³⁷.

Page 16. Line (Discussion) The data shown here also provide a physiological context and a potential mechanism for the protective role of eIF5A knockdown in anoxia⁹² and the hypothesized tumor suppressor role of eIF5A⁹³; although, we are cognizant that this may depend on the genetic background of various cellular models^{40,49,50,94}.

Comment #3. Related to the above, the most of experiments (including those in vivo) were carried out using siRNA. What is the evidence that eIF5A levels were suppressed across the course of the study? More stable depletion strategies as well as the appropriate rescue experiments should be considered.

Response. These are similar questions as Reviewer 2 that we have addressed as follows. We show new data that siRNA treatment efficiently depleted eIF5A during at least 7 days with a similar effect on Tsc2 downregulation. We also include single-site siRNA against eIF5A and obtained similar results.

New Data.

New Supp. Fig. 3g. Effect of prolong silencing eIF5A on Tsc2

Comment #4. Stronger evidence that nuclear export of eIF5A should be provided. Using eIF5A mutants which are restrained to the nucleus should be preferred over just depleting eIF5A levels, to support the authors' claims.

Response. We provide new experiments to address this question. First, treatment of cells with leptomycin B causes nuclear retention of eIF5A and prevents Tsc2 protein induction in acidotic conditions. Second, treatment with Sirt1 inhibitor Ex-527 results in nuclear retention of eIF5A and impairs Tsc2 accumulation in acidotic cells. We prefer these experiments since nuclear export-mutants of eIF5A may also affect other functions/interactions required for Tsc2 mRNA translation rendering results difficult to interpret. In addition, mapping eIF5A domains involved in nuclear retention could not reasonably be done within the limited timeframe given by the Editor. We hope the Reviewer will agree with our experimental rationale.

New Data.

New Fig. 6f-g. Effect of leptomycin B-induced nuclear retention of eIF5A on Tsc2 protein induction during anaerobic acidosis.

New Fig. 7g-h. Effect of Ex-527-induced nuclear retention of eIF5A on Tsc2 protein induction during anaerobic acidosis.

New Text.

Page 11. In agreement with this, treatment with leptomycin B, which inhibits eIF5A nuclear export^{71,72} (**Fig. 6e**) prevented anaerobic acidosis-induced cytoplasmic export of Tsc2 mRNA (**Fig. 6f**), leading to a reduction in steady-state Tsc2 protein levels (**Fig. 6g**).

Page 12. Likewise, Sirt1, but not Sirt2, inhibition, prevented eIF5A cytoplasmic localization (**Fig. 7e**), reduced eIF5A association with translating ribosomes (**Fig. 7f**), and attenuated Tsc2 protein induction (**Fig. 7g**) and mRNA cytoplasmic export (**Supplementary Fig. 6d**) under hypoxia acidosis conditions.

Comment #5. TSC2-dependent suppression of mTOR signaling under acidosis was reported previously (PMID:21738705) and this study should be referenced by the authors. It is also surprising that authors did not exclude alternative mechanisms of TSC-dependent mTOR inactivation in acidosis (e.g. AMPK), which I thought was warranted to support their conclusions.

Response. We apologize for having omitted PMID: 21738705; this reference is cited in the corrected version of the paper. We have also examined the role of AMPK in acidosis. Results show that AMPK is not implicated in Tsc2 regulation during acidosis, consistent with other groups that have shown a suppressive effect of extracellular acidity on AMPK (PMID: 27531309, 30404151).

New Data.

New Supplementary Fig. 5e. Role of AMPK on Tsc2 during anaerobic acidosis.

New Text.

Page 10. We found that AMPK had no effect on Tsc2 phosphorylation in acidosis, consistent with previous reports that acidosis inhibits AMPK activity^{68,69} (**Supplementary Fig. 5e**).

Page 9. Interestingly, Tsc2 mediated suppression of mTOR under acidotic conditions has been reported⁶⁴.

Comment #6. Term “metabolic depression” seems inappropriate. The authors should consider using “metabolic reprogramming” or a similar term.

Response. We use the term “metabolic depression” in keeping with terminology used in other papers studying the effect of extracellular low pH on metabolic activity. We hope that the Reviewer will understand why we use this terminology.

Comment #7. Figure 7 – Sirt1 depletion (KO or KD) condition would be useful to support the Sirt1 inhibitor (Ex-572) data

Response. As requested by the Reviewer, we have examined the effect of siRNA-mediated silencing of Sirt1 on eIF5A acetylation status. Results show that silencing Sirt1 has similar effect on eIF5A as treatment with Ex-572.

New Data

New Fig. 7d. Effect of silencing Sirt1 on eIF5A acetylation

New Text.

Page 12. Page 12. Sirt1 and Sirt2 are two major eIF5A deacetylases in cells and in enzyme/substrate in vitro assays⁴¹ that are regulated by NAD⁺, a principal product of fermentation^{75,76}. We found that Sirt1 and Sirt2 specifically deacetylates eIF5A during anaerobic acidosis and in neutral conditions, respectively (**Fig. 7c-d, Supplementary Fig. 8e**).

Comment #8. Fig 2.i,j) is n=6 sufficient to draw these conclusions? What is the power of this study? Related to the above, these studies were carried out with siRNA. What is the evidence that eIF5A was still depleted during the course of the experiment?

Response. Please see Comment #3 for new data that siRNA treatment depletes eIF5A during at least 7 days with a similar effect on Tsc2 downregulation. Statistically-relevant differences are observed after 7 days in the tumor assay. The power of this experiment is in-line with recommendations from SCCC animal facilities, our animal welfare protocol and was as described in our NCI grant. Briefly, the Statmate program for binomial test (two-sided) and $\alpha=0.05$, the number of animals per group is n=6 to achieve power of at least 0.8 (1- β) with effect size of 0.60 and higher.

Comment #9. Statistical analyses are not clearly described (there is no description which tests were used in the figures). Some additional issues with statistical analyses were also observed. For instance, it is more appropriate to use SD than SEM for e.g. densitometry measurements. It is also not clear why is densitometry used in some cases but not the others. Moreover, if controls were done in triplicate the authors should propagate SD values and present them on graphs.

Response. Description of the statistical test used for each figure is now added to the figure legends. We have included densitometry with SD values instead of SEM to the corresponding supplemental figures for relevant western blots. For some of the figures the data was normalized to the control as described in figure legends.

Comment #10. Only evidence of metabolic reprogramming comes from the levels of enzymes involved in the metabolic pathways and not direct measurements of metabolites in these pathways. Therefore, the authors should either perform adequate measurements of metabolites or tone down their claims.

Response. As per the request of Reviewer #2 and #3, we have removed several panels of Figure 3 and tone down claims of metabolic reprogramming.

Comment #11. It is not clear which of the eIF5A isoforms is implicated in the observed phenomena (eIF5A1 or 2)?

Response. The MATRIX analysis specifically identified eIF5A1 but not eIF5A2 as the translation factor that accumulates in heavy polysome during anaerobic acidosis; this is why we focused our study on eIF5A1. Based on the manufacturer reagents, the siRNA specifically targeted eIF5A1 and not eIF5A2. This was validated by silencing eIF5A1 that had no effect on eIF5A2 protein levels and vice-versa.

New Data

New Supplementary Fig. 2e. Specificity of reagents to eIF5A1.

New Text

Page 25. Supp. Material and Methods. The eIF5A1 antibodies and siRNAs used in this study were specific to eIF5A1 and not eIF5A2 as described by the manufacturer and validated in (Supplementary Fig. 2e).

Page 6. MATRIX identified eIF5A1 as a translation factor enriched in polysomes of hypoxic acidic cells (dark blue columns) compared to normoxic neutral pH (**Fig. 1b**) and hypoxic neutral pH (**Fig. 1c**) conditions (eIF5A1 will be hereafter referred to as eIF5A^{39,40}).

Page 13. This axis may be broadly applicable to other physiological conditions and/or compound treatments known to activate Sirt1^{78,79}, suppress mTORC1^{80,81} and may potentially involve eIF5A2 in specific settings.

Comment #12. In supplementary figure 1a the colors of lines are very similar and hard to distinguish. I appreciate that the colors are chosen to represent “hypoxia neutral pH” and “hypoxia acidosis” states throughout the manuscript, but in this particular case the authors are encouraged to e.g. use a broken line for one of the conditions.

Response. We have changed the color code across the manuscript to better highlight various conditions.

Comment #13. It should be clarified what was the precise criteria for the distinction between what is referred to as oligosomes vs. polysomes. There is already enough confusion in the field, so the authors are encouraged to use simpler terms such as light and heavy polysomes.

Response. We have relabeled oligosomes vs. polysomes to light and heavy polysomes across the text and figures.

Comment #14. In figure 5b the number of replicates is not indicated.

Response. We have indicated the number of replicates in Fig. 5b

Comment #15. In supp. figure 6A (WI-38 normoxia neutral pH condition) DAPI does not appear to correspond to eIF5A staining. Authors should consider showing panel overlays.

Response. As show below, the DAPI corresponds to eIF5A staining.

Comment #16. The authors state “This contrasts with the role of Sirt2 as a major deacetylase of eIF5A under basal conditions” indicating the role of Sirt2 as a deacetylase of eIF5A in basal conditions. However, in Supplementary Fig 6. c) with the use of a Sirt2 inhibitor (AGK2) there is no marked rise in eIF5A acetylation.

Response. We agree with the Reviewer and have reproduced experiments of Supplementary Fig 8e that shows an increase in eIF5A acetylation in basal conditions in cells treated with a Sirt2

inhibitor, as previously reported by others (PMID: 22771473). Inhibition of Sirt1 had a more modest effect on eIF5A acetylation status under basal conditions.

New Supplementary Fig. 8e

Comment #17. Work from Tauc's group shows that in kidney suppression of eIF5A activity (by targeting hypusination) prevents cell death under anoxia by silencing mitochondrial functions. This work should be commented on.

Response. We have cited this work.

New Text

Page 14. The data shown here also provide a physiological context and a potential mechanism for the protective role of eIF5A knockdown in anoxia⁹² and the hypothesized tumor suppressor role of eIF5A⁹³; although, we are cognizant that this may depend on the genetic background of various cellular models^{40,49,50,94}.

Reviewers' comments:

Reviewer #1 (Remarks to the Author):

The authors have now addressed all of the points raised in my report.

Reviewer #2 (Remarks to the Author):

The Rely to the rebuttal letter

The authors have made significant improvements to address the comments; however, in my opinion there are some issues that have be addressed for the manuscript to be acceptable for publication.

Comment #1.

"we include Source Tables with raw MS datasets in the revised version of the manuscript."

Where? I can't find the source table? you only deposited the raw. It is essential to include table with complete list of identified proteins and their corresponding intensities and the relative fold change between conditions.

Also, it is not clear for the reader how fig1b and 1c, sup. fig2 b and c are generated? the proteins shown in the figures are the only proteins identified using the MATRIX approach? If yes how and why you discarded the unshown hits? Again, without a supplementary table with raw data, it is very difficult to interpret the results (same applies for TMT-pSILAC).

"For the MATRIX, we selected eIF5A since it was the only translation factor that displayed an increase in translating ribosomes in hypoxic/acidotic cells compared to both normoxic/neutral and hypoxic/neutral "

Authors have clearly shown that eIF5A is pH-regulated translation factor. However, using the MATRIX approach with single replicate without any statistical evidence is not possible to state or hypothesis that eIF5A is the only translation factor with such property. For example, changing arbitrary fold cut-off to >zero or 0.5, additional candidates will also fit the criteria as eIF4B, eIF4G2, eIF5 and eIF2 α (based on fig.1b and c), or eIF2 α , eIF4B(if supp.fig2 b and c are included in the assessment). For the current manuscript to accepted for publication, The Authors need either to include additional replicates for their experiments (for both the MATRIX and TMT-pSILAC) or to perform the proper statistical analysis to evaluate the noise level in their datasets.

"As we successfully validated the MATRIX"

This is a very generous interpretation of validation. Maybe you can be more specific as I can only see the western blot validation for eIF5A, eIF3d and eIF3k.

Comment #2.

"We believe that PEAKS performs in a superior manner for SILAC data"

This reviewer was interested to know why there was inconsistency. It's fair enough if you found one approach superior. However, this statement reinforces the earlier point that a robust statistical evaluation of the data is necessary prior to making any biological interpretation. That is absolute minimum when presenting -omics data.

I hope that the authors will find these comments helpful. once these comments are addressed, I believe that manuscript will be suitable for publication.

Reviewer #4 (Remarks to the Author):

In this revised manuscript that authors have addressed the criticisms and comments raised regarding the original submission. I believe that the changes are reasonable.

That being said, I have a few comments on the paper:

1. the massive polysome loss in hypoxia acidosis conditions (Fig 2a) indicate a defect in translation initiation. This could in part account for the de-enrichment of initiation factors from the large polysomes. These initiation factors were most likely associated with the subset of initiating ribosomes on the polysomal mRNAs (mRNAs with multiple elongating ribosomes and one or a few initiating ribosomes). The relative enrichment of eIF5A could be due to this loss of initiating ribosomes leading to a relevant increase in elongating ribosomes and apparent increase in elongation factors on the polysomes.

2. The role for eIF5A in translation elongation is well-supported both genetically and biochemically. The same is not true for the purported function of the protein in mRNA transport. While the authors' data is consistent with their model, I remain skeptical that the observed effects are due to a role of eIF5A in mRNA transport. Could inhibition of eIF5A directly impair expression of the transporter and thereby indirectly affect mRNA export?

Minor comments:

1. Line 71: references 53 and 54 do not support the authors' statement. I believe they cited the wrong papers

2. Figure 2a has the oligosome nomenclature; should change to light polysomes

3. Figure 4b: in the translational field "delta TE" has acquired a specific meaning related to ribosomal profiling experiments (Ingolia et al). As the authors' experiments are measuring the relative polysome association of specific mRNAs, I believe a different term should be used.

Response to Reviewer #2

We are pleased that the Reviewer stated that we have made significant improvements to address the original comments. Below is our detailed point-by-point response to every remaining comment of the Reviewer.

Comment #1:

The authors have made significant improvements to address the comments; however, in my opinion there are some issues that have be addressed for the manuscript to be acceptable for publication

“we include Source Tables with raw MS datasets in the revised version of the manuscript.” Where? I can’t find the source table? you only deposited the raw. It is essential to include table with complete list of identified proteins and their corresponding intensities and the relative fold changes between conditions. Also, it is not clear for the reader how fig1b and 1c, sup. fig2 b and c are generated? the proteins shown in the figures are the only proteins identified using the MATRIX approach? If yes how and why you discarded the unshown hits? Again, without a supplementary table with raw data, it is very difficult to interpret the results (same applies for TMT-pSILAC). “For the MATRIX, we selected eIF5A since it was the only translation factor that displayed an increase in translating ribosomes in hypoxic/acidotic cells compared to both normoxic/neutral and hypoxic/neutral “Authors have clearly shown that eIF5A is pH-regulated translation factor. However, using the MATRIX approach with single replicate without any statistical evidence is not possible to state or hypothesis that eIF5A is the only translation factor with such property. For example, changing arbitrary fold cut-off to >zero or 0.5, additional candidates will also fit the criteria as eIF4B, eIF4G2, eIF5 and eIF2α (based on fig.1b and c), or eIF2α, eIF4B(if supp.fig2 b and c are included in the assessment). For the current manuscript to accepted for publication, The Authors need either to include additional replicates for their experiments (for both the MATRIX and TMT-pSILAC) or to perform the proper statistical analysis to evaluate the noise level in their datasets. “As we successfully validated the MATRIX” This is a very generous interpretation of validation. Maybe you can be more specific as I can only see the western blot validation for eIF5A, eIF3d and eIF3k.

Response:

We sincerely apologize for our lack of clarity. We have made sure the unfiltered list of translation factors identified by MATRIX is now clearly visible in the source data file. We explain here, and in the revised manuscript the rationale and criteria we established *a priori* that led us to focus on eIF5A as the proof-of-concept candidate. We agree absolutely with the Reviewer that eIF5A is most likely not the only translation factor activated by hypoxia acidosis. It was not our intention to suggest this, and we apologize if it came across that way.

In addition to including the source file, we have made significant changes to the text (results, discussion and M&M) as well as including a schematic diagram in the Supp. Figures to clarify our strategy.

Specifically, for Fig. 1b, 1c, S2b and S2c, we focused on canonical translation factors, of which MATRIX identified 51 by ≥ 2 unique peptides. In total, MATRIX identified 4017 proteins by ≥ 2 unique peptides. We applied our pre-established criteria to the 51 translation factors:

1. Candidates have to be detected across all fractions. 28 out of 51 passed this filter.
2. For our primary comparison between hypoxia acidosis and basal conditions, we used the ratio of protein abundance in polysome over ribosome-free fractions as the primary readout for translational engagement. We previously established the effectiveness of this ratio (PMID: 29298419). The standard 2-fold cut-off was chosen. Nine translation factors passed this filter: eIF4G1, eEF2, PABPC1, eIF4G2, ETF1, eIF5A, eIF4B, eIF4H, and eIF2A.
3. Outlier identification using the interquartile range method excluded eIF4B, eIF4H, and eIF2A as far outliers.
4. Next, we eliminated candidates that are also activated by hypoxia alone, with resulted in the exclusion of eIF4G1, eEF2, PABPC1, eIF4G2, and ETF1.
5. Finally, to improve our confidence that candidates are indeed associated with actively translating ribosomes, we used the ratio of protein abundance in polysome fractions over 40S/60S/monosome fractions as a secondary readout to eliminate candidates that may be stalled at the initiation step of translation. We maintained consistency by using the standard cut-off of 2-fold increase compared to both basal and hypoxia alone conditions.

eIF5A was the only candidate that passed these criteria, and hence was pursued as the focus of our current study, especially since we noticed a robust phenotype consistent with our main hypothesis by silencing this translation factor. We have included and referenced an individual excel source file, termed `MATRIX_sourcefile`, that includes all proteins found and the filtering, calculations used, as well as for `TMT-pSILAC_sourcefile`.

We have added new text to clarify our interpretations. In the current study, we chose the widely-used cut-off of 2-fold increase to maintain consistency with previous studies from our group (29298419) as well as others (19955088, 21906983). We agree that reducing the stringency (to 1.5-fold compared to basal conditions, for example) would result in additional candidates being included, e.g. eIF5B, eIF2S1 and eEF1G. However, none of these candidates showed preferential activation in hypoxia acidosis when compared to hypoxia alone. We acknowledge that there may be many more hypoxia acidosis-activated targets worth pursuing. In this study, we focused on the top candidate based on our analysis as described above.

We are also cognizant of the fact that the aforementioned analysis was focused on known and established translation factors. As an unbiased approach screen, MATRIX identified a number of protein factors that matched the criteria described above, yet have no reported roles in protein synthesis. This is an area of active investigation by us, which we hope the Reviewer will agree warrants additional independent studies that lie beyond the scope of our current manuscript.

The purpose of the MATRIX screen was to identify candidates for further pursuit. Although the MS measurements were performed on a single pooled replicate, we focused on the top candidate eIF5A as our datasets were consistent with our main hypothesis: “*cells activate a unique translational program to produce key proteins required to suppress metabolism and preserve genomic integrity*”. The role of eIF5A was confirmed using multiple independent approaches, including 1) immunoblot confirmation of its distribution across ribosome density fractions, 2) subcellular localization, 3) hypoxia acidosis-specific effects of its knockdown on key phenotypes of hypoxia acidosis including proliferation, DNA replication, ATP utilization, transcription, DNA damage *etc.* We feel that the use of multiple independent approaches provides unique advantages for establishing the biological significance of eIF5A. Notably, this approach circumvents the inherent limitations of any single technique, and it reduces confirmation bias. Given that the Reviewer agrees the “authors have clearly shown that eIF5A is pH-regulated translation factor”, and that we have confirmed the biological relevance of MATRIX-identified eIF5A using numerous independent approaches, we sincerely hope the Reviewer will concur that the inclusion of additional MATRIX replicates would not affect the conclusions of the paper, especially considering the significant time and financial investments required for mass spectrometry.

We chose representative targets that span the spectrum of hypoxia acidosis-induced changes in activity. Specifically, eIF5A was chosen as a target that was activated, eIF3k as a target that was down-regulated in terms of activity, and eIF3d was chosen as a representative factor whose translational activity was maintained (neither up- nor down-regulated) by hypoxia acidosis. This point is indicated in Fig. 1d. We have removed the term “validated or validation” across the paper.

New Supplementary Fig. 2b (each filter step is included in different tabs of the source file)

New text:

Results section: “MATRIX successfully identified 51 canonical translation factors (each by at least 2 unique peptides). We applied a series of stringency criteria to identify the most promising candidate(s) (**Supplementary Fig. 2b, MATRIX_sourcefile**). We narrowed down on candidates (28 out of 51) that were detected across all fractions. Next, using the ratio of protein abundance in polysome fractions over ribosome-free fractions as our primary readout for translational engagement, we further narrowed our attention to those translation factors that exhibited at least a 2-fold increase in hypoxia acidosis, in-line with our previous studies²⁰ and a cut-off used by other groups^{46,47}, compared to basal conditions. Following outlier removal using the interquartile range method, we eliminated candidates that were also found to be activated by hypoxia alone. Finally, we applied a 2-fold cut-off to our secondary readout i.e. the ratio of protein abundance in polysome fractions over monosome (40S/60S/80S) fractions, to eliminate candidates that may be stalled at the initiation step of translation, and to improve our confidence that candidates are indeed associated with actively translating ribosomes. Translation factors that satisfied all the aforementioned criteria were pursued as the focus for further investigation.

Results section: Reducing the stringency (to 1.5-fold compared to basal conditions, for example) resulted in additional candidates being included, e.g. eIF5B, eIF2S1 and eEF1G. However, neither of these candidates showed preferential activation in hypoxia acidosis when compared to hypoxia alone.

Discussion section: “We point out that even though the current study is focused on eIF5A, our unbiased MATRIX screen revealed additional candidate proteins that may be activated by hypoxia acidosis, including some that are not known to participate in protein synthesis. Further studies are warranted to investigate the role of additional hypoxia acidosis-activated translation factors and regulators.”

Materials and Methods: We filtered mass spectrometry results to peptides found with greater than one unique peptide. For translation factor enrichment we filtered results to canonical translation factors that were detected in all conditions (normoxia neutral, hypoxia neutral, and hypoxia acidosis) and in every fraction (free, monosome, polysome). Ratio of polysome to free was determined in every conditions to estimate translational engagement. We considered a greater than 2-fold increase in acidosis compared to the other conditions as enriched in acidosis. We eliminated extreme outliers using the interquartile range method to remove potentially noisy data. To improve our confidence that candidates are indeed associated with actively translating ribosomes, we used the ratio of protein abundance in polysome fractions over 40S/60S/monosome fractions as a secondary readout to eliminate candidates that may be stalled at the initiation step of translation. We maintained consistency by using the standard cut-off of 2-fold increase compared to both basal and hypoxia alone conditions (MATRIX_source file).

Comment #2.

“We believe that PEAKS performs in a superior manner for SILAC data” This reviewer was interested to know why there was inconsistency. It’s fair enough if you found one approach superior. However, this statement reinforces the earlier point that a robust statistical evaluation of the data is necessary prior to making any biological interpretation. That is absolute minimum when presenting -omics data. I hope that the authors will find these comments helpful. once these comments are addressed, I believe that manuscript will be suitable for publication.

I hope that the authors will find these comments helpful. once these comments are addressed, I believe that manuscript will be suitable for publication.

Response:

We sincerely apologize for the confusion relating to the use of different analysis software for MATRIX and TMT-pSILAC. MATRIX was performed first, and was analyzed using PEAKS software. For TMT-pSILAC, the combination of TMT and SILAC (together with sample fractionation) yields a more complex data set compared to MATRIX, in which TMT was not employed. At the time we obtained the TMT-pSILAC data, PEAKS was not optimized to support the analysis of TMT data, which is the reason Proteome Discoverer was used instead for TMT-pSILAC. Internal comparisons between the two softwares found them to be highly comparable in terms of overall peptide identification and quantification, and also specifically for our targets-of-interest Tsc2 and Sirt1. We modified the text to clarify these points.

New text:

Results section: Here, we chose 1.5-fold as the threshold for enhancement based on the average induction of glycolytic enzymes in hypoxia-neutral compared to normoxia-neutral⁵⁸⁻⁶⁰ (**Supplementary Fig 4b, TMT-pSILAC_sourcefile**). Well-characterized candidates uncovered by TMT-pSILAC with available high-quality reagents and that displayed various fold-enrichment above a threshold of 1.5-fold^{61,62} were tested by immunoblot analysis (**Fig. 3d, Supplementary Fig 4c**).

Materials and Methods: Data was corrected for equal protein concentration used for TMT-labeling and detection and cell number before analysis of fold-change (correction factor HA: 0.650154799; HN: 0.80804954). Results were further filtered for proteins that were detected in every conditions (NN, HN, HA). Ratio of HA/NN and HA/HN was determined and ratios greater than 1.5 (based on induction of glycolytic enzymes in HN vs. NN) was considered induced in acidosis (TMT-pSILAC_sourcefile).

New Supplementary Fig 4b (each filter step is included in different tabs of the source file)

b

TMT-pSILAC Filtering Criteria

Data corrected for cell number and protein concentration in hypoxia-neutral and hypoxia-acidotic conditions (Tab #1)

Candidates must be detected across all conditions (NN, HN, HA) = 3533 retained (Tab #2)

Induction cut-off, >1.5, determined based on average induction of anaerobic-glycolytic enzymes in hypoxia neutral compared to normoxia neutral (Tab #3)

Ratio of HA/NN and HA/HN was calculated; induction of >1.5 in one condition but not the other was considered acidosis specific to that condition (HA/HN specific = 250, HA/NN specific = 92) (Tab #4 and #5)

Targets that showed a >1.5 induction in acidosis when compared to both hypoxia and normoxia neutral conditions were considered acidotic-specific candidates = 244 (Tab #6)

Response to Reviewer #4

We are grateful to the Reviewer for stating that we have addressed the criticisms and comments raised regarding the original submission in a reasonable manner. Below is our detailed point-by-point response to every comment of the Reviewer.

Comment #1:

In this revised manuscript that authors have addressed the criticisms and comments raised regarding the original submission. I believe that the changes are reasonable.

That being said, I have a few comments on the paper:

The massive polysome loss in hypoxia acidosis conditions (Fig 2a) indicate a defect in translation initiation. This could in part account for the de-enrichment of initiation factors from the large polysomes. These initiation factors were most likely associated with the subset of initiating ribosomes on the polysomal mRNAs (mRNAs with multiple elongating ribosomes and one or a few initiating ribosomes). The relative enrichment of eIF5A could be due to this loss of initiating ribosomes leading to a relevant increase in elongating ribosomes and apparent increase in elongation factors on the polysomes.

Response:

The Reviewer is right in stating that there is an important loss of polysomes in hypoxic acidotic conditions that is likely a consequence of a defect in the canonical translation initiation pathway(s), a possible consequence of Tsc2-mediated inactivation of mTOR (Figure 4). This may lead to a relative increase of elongation factors, amongst other participants of hypoxic/acidotic translation, in the polysome fractions. We have added a sentence in the discussion to address this comment. Nonetheless, we kindly remind that we confirmed the increased eIF5A activation and eIF5A-dependence of hypoxia-acidotic cells compared to their basal counterparts using various independent approaches throughout the entire manuscript e.g. knockdown experiments, subcellular localization etc.

New Text:

Discussion: In addition, the important loss of polysomes observed during hypoxia/acidosis likely as a consequence of Tsc2-mediated mTOR inhibition may also cause a relative enrichment of polysomic eIF5A, amongst other translation factors, detected by MATRIX.

Comment #2:

The role for eIF5A in translation elongation is well-supported both genetically and biochemically. The same is not true for the purported function of the protein in mRNA transport. While the authors' data is consistent with their model, I remain skeptical that the observed effects are due to a role of eIF5A in mRNA transport. Could inhibition of eIF5A directly impair expression of the transporter and thereby indirectly affect mRNA export?

Response:

We completely agree with the reviewer that a role for eIF5A in translation elongation is well-established in the literature. We tested many different assays and found that extracellular acidosis activated Sirt-1-dependent nuclear export of eIF5A in a leptomycin-sensitive manner. Our datasets are, thus, in-line with a substantial body of excellent studies that have highlighted a role of eIF5A in nuclear transport in cells responding to various stimuli (20501948, 8596953, 10790432). Indeed, previous reports have shown that eIF5A is a nuclear/cytoplasmic trafficking protein that engages in CRM1 (exportin 1)/leptomycin B-dependent nuclear export of various proteins and mRNAs (20501948, 10381392, 10339570). Specifically, we show that activated eIF5A undergoes CRM1/leptomycin B-dependent nuclear export in a complex with Tsc2 mRNA, amongst other target mRNAs, consistent with a published paper that identified a role of eIF5A in mRNA export (20501948) in cells on stimuli. Surely, these results do not exclude the possibility that eIF5A plays a role in translation elongation for many proteins during anaerobic acidosis.

As requested by the Reviewer, we have performed a western blot of known eIF5A transporters (CRM1 and Xpo4). As shown below, silencing eIF5A had no effect on levels of these two transporters (CRM1/Xpo4). We have modified the text to better explain our data in the context of the literature.

New Data: Effect of silencing of eIF5A on CRM1 and Xpo4.

New text:

Results: In stark contrast, eIF5A depletion had no effect on nuclear export and polysome engagement of eIF5A-independent mRNAs (**Fig. 6b, Fig 6d, Supplementary Fig. 7d**) or on the levels transporters CRM-1⁸¹⁻⁸³ and Xpo4⁸⁴ involved in eIF5A nuclear export (**Supplementary fig. 7e**).

Discussion: Our data suggest that a principal role of activated eIF5A in acidotic cells is to export target mRNAs from the nucleus to the cytoplasm for efficient translation, a function of eIF5A that has been observed by other groups^{82,102} in cells responding to stimuli. Our data do not exclude that eIF5A may be involved in translation of polyproline motifs of other proteins under acidotic conditions⁸⁶

Minor Comments

Comment #1

Line 71: references 53 and 54 do not support the authors' statement. I believe they cited the wrong papers.

Response:

We apologize for this mistake and have change the references.

Comment #2

Figure 2a has the oligosome nomenclature; should change to light polysomes

Response:

We have changed the term “oligosome” to “light polysome”

Comment #3

Figure 4b: in the translational field “delta TE” has acquired a specific meaning related to ribosomal profiling experiments (Ingolia et al). As the authors' experiments are measuring the relative polysome association of specific mRNAs, I believe a different term should be used.

Response: Wherever possible, we have changed the term to “TE^{HA}/TE^{NN}”. We used the “TE” nomenclature to maintain consistency with our previous publications (e.g. PMID: 26854219). We have added a sentence to explicitly state our definition of TE, and point out the difference with its usage in other studies.

New text:

Results section. We define translation efficiency (TE) as the ratio of mRNA abundance in polysome fractions to that in free/monosome fractions. We note the difference between this definition from its usage in other studies e.g. ribosome profiling experiments⁶³.

REVIEWERS' COMMENTS

Reviewer #2 (Remarks to the Author):

The authors have answered the outstanding questions and the manuscript is now acceptable.

Reviewer #4 (Remarks to the Author):

In this revised manuscript, the authors have attempted to address my concerns with the previous submission. While I remain very skeptical that eIF5A is "activated" during hypoxic acidosis in the sense that eIF5A is clearly active under basal conditions to promote translation. Moreover, I am very skeptical that the primary role for eIF5A is to promote transport of the tsc2 and other mRNAs under these conditions. That being said, I will agree that the authors have provided data consistent with this hypothesis (depletion of eIF5A correlates with loss of tsc2 mRNA transport) and the authors have addressed all of the minor concerns I raised in regard to the previous submission.

Contrary to the authors' contention, I respectfully disagree that other "excellent" studies have highlighted a role for eIF5A in mRNA transport, I have doubts about the conclusions from all of these studies. The only biochemically verified function of eIF5A is to promote translation elongation by facilitating peptide bond formation. This translational function supports the synthesis of every peptide bond, consistent with eIF5A binding in the E site of the ribosome. Some peptide bonds show a greater dependency on eIF5A (like polyproline), but many non-proline motifs are also highly dependent on eIF5A for peptide bond synthesis.

Unfortunately, I was not clear in one of my comments regarding the previous version of this paper. I did not propose that the expression of the eIF5A transporter was altered under hypoxic acidosis conditions. Rather, I meant to propose that eIF5A promotes the translation of the mRNA encoding the true and unknown tsc2 mRNA transport factor.

Finally, I believe it is also worth noting that the authors' proposed model for eIF5A does not support their initial hypothesis about a unique translational program under hypoxic acidosis conditions. Instead, their final model is regulation at the level of mRNA transport not translation. They have not demonstrated a translational re-programming as indicated in the title and abstract, but instead propose an mRNA transport re-programming.

Response to Reviewers.

Reviewer 4.

Comment #1. “I remain very skeptical that eIF5A is “activated” during hypoxic acidosis in the sense that eIF5A is clearly active under basal conditions to promote translation. Moreover, I am very skeptical that the primary role for eIF5A is to promote transport of the tsc2 and other mRNAs under these conditions. That being said, I will agree that the authors have provided data consistent with this hypothesis (depletion of eIF5A correlates with loss of tsc2 mRNA transport) and the authors have addressed all of the minor concerns I raised in regard to the previous submission.”

Response. It was not our intention to imply that eIF5A is inactive under basal conditions. To clarify the message, we have replaced “activated” with “adaptively engaged”, and rephrased various sentences throughout the manuscript to maintain the consistency of this message. We have modified the text as follows:

Modified Text

Page 4. This pathway requires the acidotic adaptive engagement of eIF5A, a protein that can be traced back to the last common universal ancestor (LUCA)^{33,34}, which evolved under anaerobic conditions³⁵.

Page 5. Anaerobic acidosis adaptively engages the eukaryotic translation factor 5A.

Page 8. We tested the biological implications of adaptive engagement of eIF5A for metabolic adaptation during anaerobic acidosis.

Page 12. Taken together, these results suggest that adaptive engagement of eIF5A prevents DNA damage through Tsc2 protein induction to suppress mTORC1 activity and cellular proliferation.

Page. 17. The identification of an acidosis-enriched translome and hypoxia acidosis-engaged translation factors e.g. eIF5A highlight the evolving concept of global translational reprogramming as a central mechanism of adaptation.

Comment #2. Finally, I believe it is also worth noting that the authors’ proposed model for eIF5A does not support their initial hypothesis about a unique translational program under hypoxic acidosis conditions. Instead, their final model is regulation at the level of mRNA transport not translation. They have not demonstrated a translational re-programming as indicated in the title and abstract, but instead propose an mRNA transport re-programming.

Response. In the text, we define “translatome remodelers” as factors that control protein outputs at the translational level (i.e. translation efficiency of mRNAs) in response to stimuli. These can be RNA-binding proteins, translation factors, nuclear/cytoplasmic transport factors, etc. As eIF5A regulates the translation efficiency of mRNAs, including tsc2, we believe that our

interpretation is consistent with the original definition of “translatome remodelers”. We have modified the text as follows:

Modified Text

Page 3. These translation factors operate as translatome remodelers that control protein outputs at the translational level (i.e. translation efficiency of mRNAs) to activate biochemical pathways

Comment 3. Contrary to the authors’ contention, I respectfully disagree that other “excellent” studies have highlighted a role for eIF5A in mRNA transport, I have doubts about the conclusions from all of these studies. The only biochemically verified function of eIF5A is to promote translation elongation by facilitating peptide bond formation. This translational function supports the synthesis of every peptide bond, consistent with eIF5A binding in the E site of the ribosome. Some peptide bonds show a greater dependency on eIF5A (like polyproline), but many non-proline motifs are also highly dependent on eIF5A for peptide bond synthesis.